# Sclerostin small-molecule inhibitors promote osteogenesis by activating canonical Wnt and BMP pathways

**Sreedhara Sangadala[1,2], Chi Heon Kim[2], Lorenzo M Fernandes[1,2], Pooja Makkar[3], George R Beck[1,4], Scott D Boden[2], Hicham Drissi[1,2]\*, Steven M Presciutti[1,2]\***

[1]Atlanta Veterans Affairs Medical Center, Decatur, United States; [2]Department of Orthopaedics, Emory University School of Medicine, Atlanta, United States; [3]Department of Biotechnology, Panjab University, Chandigarh, India; [4]Emory University, Division of Endocrinology, Atlanta, United States

## Abstract

**Background:** The clinical healing environment after a posterior spinal arthrodesis surgery is one of the most clinically challenging bone-healing environments across all orthopedic interventions due to the absence of a contained space and the need to form de novo bone. Our group has previously reported that sclerostin in expressed locally at high levels throughout a developing spinal fusion. However, the role of sclerostin in controlling bone fusion remains to be established.

**Methods:** We computationally identified two FDA-approved drugs, as well as a single novel small-molecule drug, for their ability to disrupt the interaction between sclerostin and its receptor, LRP5/6. The drugs were tested in several in vitro biochemical assays using murine MC3T3 and MSCs, assessing their ability to (1) enhance canonical Wnt signaling, (2) promote the accumulation of the active (non-phosphorylated) form of β-catenin, and (3) enhance the intensity and signaling duration of BMP signaling. These drugs were then tested subcutaneously in rats as standalone osteoinductive agents on plain collagen sponges. Finally, the top drug candidates (called VA1 and C07) were tested in a rabbit posterolateral spine fusion model for their ability to achieve a successful fusion at 6 wk.

**Results:** We show that by controlling GSK3b phosphorylation our three small-molecule inhibitors (SMIs) simultaneously enhance canonical Wnt signaling and potentiate canonical BMP signaling intensity and duration. We also demonstrate that the SMIs produce dose-dependent ectopic mineralization in vivo in rats as well as significantly increase posterolateral spine fusion rates in rabbits in vivo, both as standalone osteogenic drugs and in combination with autologous iliac crest bone graft.

**Conclusions:** Few if any osteogenic small molecules possess the osteoinductive potency of BMP itself – that is, the ability to form de novo ectopic bone as a standalone agent. Herein, we describe two such SMIs that have this unique ability and were shown to induce de novo bone in a stringent in vivo environment. These SMIs may have the potential to be used in novel, cost-effective bone graft substitutes for either achieving spinal fusion or in the healing of critical-sized fracture defects.

**Funding:** This work was supported by a Veteran Affairs Career Development Award (IK2-BX003845).

**\*For correspondence:**
hicham.drissi@emory.edu (HD);
prescius33@gmail.com (SMP)

**Competing interest:** The authors declare that no competing interests exist.

## Editor's evaluation

This manuscript examines the use of small molecule sclerotin inhibitors as alternatives to currently used biologicals such as BMP for interventions such as spinal fusion. The manuscript has important clinical significance and the strength of evidence is considered convincing.

## Introduction

The current bone grafting options available to aid orthopedic surgeons in either the repair of fractures with significant bone loss or in the achievement of spinal fusions continue to have serious limitations. Thus, autologous iliac crest bone graft (ICBG) remains the gold standard in both settings, but its harvest has been associated with long-term donor site pain in up to 25% of patients and its use in the posterolateral spine has a non-union rate nearing 40% (*Noshchenko et al., 2014*). A variety of alternative bone graft options are currently available (i.e. allograft, demineralized bone matrix); however, none have been clinically proven to be suitable as effective autograft substitutes (*Boden, 2000*). Based on this clinical need, we sought to develop a novel osteogenic treatment strategy that targets the established osteogenic capacity of the canonical Wnt signaling to safely and cost-effectively promote spinal fusion.

The canonical Wnt signaling pathway stands out as a promising pathway to target for bone regenerative therapies given that it is central to the development and homeostasis of bone and because it has complex crosstalk with canonical BMP signaling. Upon the binding of Wnt ligands to the LRP5/6 (low-density lipoprotein receptor-related protein 5/6) receptor, β-catenin accumulates and translocates into the nucleus where it forms complexes with TCF/Lef1 (T-cell factor/lymphoid enhancer factor) transcription factors (*Baron and Rawadi, 2007*). This complex then initiates the transcription of various osteogenic target genes. In osteoblasts, these genes enhance proliferation, expansion, and survival, significantly increasing osteogenesis (*Boden et al., 1995*; *Nusse, 2005*). The regulation of intracellular canonical Wnt signaling is ensured physiologically by the Wnt antagonist, sclerostin, which, upon binding the LRP5/6 receptor, disrupts the interaction between extracellular Wnts and their receptor. This potent inhibitor of the Wnt/β-catenin pathway is secreted primarily by osteocytes and functions to inhibit bone formation (*Li et al., 2005*). Sclerostin inhibition therefore remains an attractive target for novel anabolic therapeutics (*Gamie et al., 2012*; *Suen and Qin, 2016*).

Most of the previously explored sclerostin-blocking strategies have focused on systemic delivery of anti-sclerostin monoclonal antibodies (mAbs). Anti-sclerostin mAbs have been shown to enhance bone healing in ovariectomized rats and bone mineral density in humans by increasing bone formation and mass due to enhanced osteoblast function (*Suen et al., 2014*; *McDonald et al., 2012*; *Recker et al., 2015*). Systemic delivery is far from ideal in local bone-healing applications like fractures and spinal fusions, however. Systemic administration often requires relatively high and sustained dosing regimens to obtain the desired therapeutic effect in the local target environment. These non-local delivery strategies also raise concerns over unnecessary systemic exposure and the potential for off-target side effects.

Building upon the proven success of systemic anti-sclerostin mAbs to enhance bone formation, we utilized a validated in silico strategy to computationally identify multiple locally deliverable small-molecule drugs with the ability to disrupt the interaction between extracellular sclerostin and LRP5/6. We hypothesized that using a small drug-like molecule to selectively prevent sclerostin from binding to LRP5/6 would enhance intracellular Wnt signaling and the efficacy of endogenous BMPs to promote osteogenesis. Herein, we show that treatments with these sclerostin small-molecule inhibitors (SMIs) resulted in significant enhancement of both canonical Wnt and BMP signaling in vitro. We also explore the possible mechanisms by which these SMIs enhance both canonical Wnt and BMP signaling, showing that crosstalk between these two pathways is partially mediated by the effect of GSK3b on the duration of Smad1 phosphorylation. We also show that these SMIs can significantly increase de novo bone formation as standalone osteogenic agents in vivo, both in a biologically stringent subcutaneous rat ectopic model and in a posterolateral spine fusion model in rabbits. Collectively, our reported findings constitute a first step toward exploring the potential use of anti-sclerostin SMIs as a novel cost-effective biological bone graft-enhancing strategy for future clinical applications such as fracture repair or spinal fusion.

## Methods

### In silico drug design

Sclerostin, belonging to the DAN protein family, has a cysteine knot that divides its intervening sequences into three loops, with the first and the third loops protruding to either side of the central knot. The loop 2 region, specifically, is the known binding site for functional anti-sclerostin antibodies

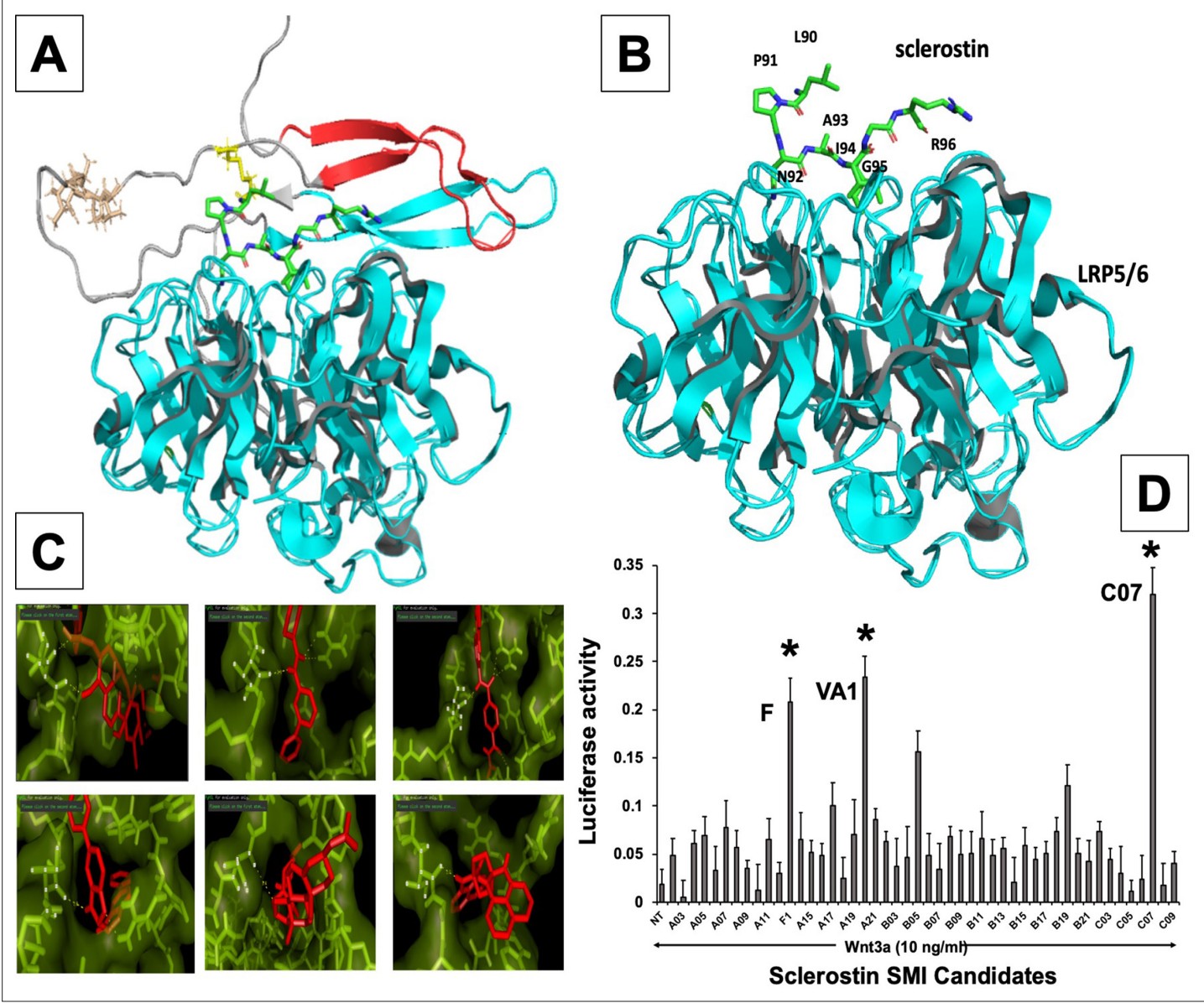

**Figure 1.** In silico design of sclerostin small-molecule inhibitors (SMIs). Structural representation of sclerostin (PDB ID 2K8P) is shown (**A**). Three loops of sclerostin are color coded (red for loop 1, gray for loop 2, and cyan for loop 3). Also, the residues from loop 2 that are targeted for inhibitor selection are highlighted, including Leu 90, Pro 91, Asn 93, Ala 93, and Ile 94, which is shown in relation to the structure of the β-propeller domain of LRP6 (PDB ID 3SOV) (**B**). Candidate compounds were next selected in silico against the loop 2 region (91st to 95th amino acid position) and the cystine knot region (Cys 85 and Cys 143) of sclerostin; their poses are shown (**C**). Panel (**D**) shows the selection of the most effective compounds in vitro, with compounds F1, VA1, and C07 being the most potent sclerostin SMIs tested. All three were able to effectively enhance Wnt3a–induced reporter activity at suboptimal dose of Wnt3a. Treatment of transfected cells with compounds alone (without Wnt3a protein) did not show any significant induction of reporter activity.

The online version of this article includes the following source data and figure supplement(s) for figure 1:

**Source data 1.** Relative luciferase activity of small-molecule screening.

**Figure supplement 1.** Optimization of suboptimal Wnt concentration in Wnt activity screening assay.

(*Figure 1A and B*; *Veverka et al., 2009*). Mutational analyses have shown that mutations in the Leu-90 to Asn-103 residues of the loop 2 region resulted in the inability for sclerostin to bind to LRP5/6 (*Boschert et al., 2016*). Similarly, a mutation of single amino acids in the loop 2 region (Asp-92 or Ile-94) is known to affect the binding of sclerostin to LRP6 (*Boschert et al., 2013*). Other studies, however, have shown that a murine sclerostin mutant with Cys-84 and Cys-142 exchange (resulting in the removal of the third disulfide bond of the cystine knot) has reduced binding affinity to LRP5/6

(*Boschert et al., 2013*), suggesting that elements outside the flexible loop 2 region are also important for proper sclerostin function. Therefore, both of these regions on sclerostin were thought to provide strong candidates to target for the development of anti-sclerostin SMIs. Virtual screening results for AutoDock Vina were visualized using PyMOL (The PyMOL Molecular Graphics System, version 2.0, Schrödinger, LLC) and for FRED using Omega. Receptor-structure-focused docking was performed using the available PDB (protein data bank) nuclear magnetic resonance spectroscopy structure of sclerostin (ID 2K8P) (*Veverka et al., 2009*) to analyze binding modes and their estimated affinities. Out of the 36 conformers of this structure, conformer 1 was chosen as it was the best representative conformer in the ensemble (as mentioned in the remarks of the PDB structure of sclerostin). For this task, we used the chemical library 'Diversity Set IV' from the National Cancer Institute (NCI), which contains 1596 compounds. Corina software was used to generate 3D coordinates of the ligands in the library (3D Structure Generator CORINA Classic, Molecular Networks GmbH; *Schwab, 2010*). The translational, rotational, and vibrational entropies of protein–drug complexes were estimated using additional modules within Discovery Studio (Dassault Systèmes BIOVIA, Discovery Studio Modeling Environment, Release 2017, San Diego), including CHARMm, GLIDE, and GOLD.

## Cell culture

The murine pre-osteoblast MC3T3-E1 cell line (*Sudo et al., 1983*) and mouse C2C12 cells, as well as Dulbecco's modified Eagle's medium (DMEM), were purchased from ATCC (Manassas, VA). The bone marrow stromal cells (BMSCs) were originally isolated from murine bone marrow by centrifugation (*Camalier et al., 2013*). All cells were used within eight passages. The non-heat-inactivated fetal bovine serum (FBS) was purchased from HyClone Laboratories, Inc (Logan, UT). Pre-osteoblasts (BMSC and MC3T3-E1) were cultured in α-modified Eagle's medium (α-MEM; Thermo Scientific) supplemented with 50 U/ml penicillin, 50 mg/ml streptomycin, 2 mM L-glutamine (Thermo Scientific), and 10% FBS (Atlanta Biologicals, Lawrenceville, GA). The C2C12 cells at passages 4–7 were subcultured in T-75 cm$^2$ flasks in DMEM supplemented with 10% FBS at 37°C in 5% $CO_2$ with humidification. When the flasks reached 60–70% confluence, the cells were trypsinized and seeded in triplicate at 200,000 cells/well in a 6-well plate for quantitative real-time RT-PCR and alkaline phosphatase (ALP) assays or at 50,000 cells/well in a 12-well plate for the dual-luciferase reporter assay.

## Dual-luciferase reporter assay

### Wnt reporter activity

A Wnt-specific TCF/LEF-driven reporter plasmid was used (QIAGEN, Valencia, CA). The C2C12 cells were trypsinized and seeded in triplicate wells at 50,000 cells/well in 12-well plates on day 1. On day 2, the cells were co-transfected with the reporter construct and the renilla-luciferase control vector using SuperFect (QIAGEN) for 24 hr. A total of 1 µg of plasmids was used for co-transfection in each well, and the concentration of renilla-luciferase vector was 1/15 of the reporter plasmid. On day 3, medium was replaced with DMEM containing 2% FBS and the cells were treated with 10 ng/ml of Wnt3a and ±10 µM of the sclerostin-interacting compound. On day 4, the luciferase activities were measured in 20 µl of cell-lysate using the dual-luciferase assay system (Promega, Madison, WI) with a luminometer (LumiCount; Packard Bioscience, Meriden, CT) following the manufacturer's instructions. The luciferase activity was expressed as relative units of luciferase (RUL; a ratio of firefly luciferase to renilla luciferase activity).

### BMP reporter activity

The BMP-specific Smad1-driven 9 × GCCG (a consensus binding sequence for Smad1) reporter plasmid was kindly provided by Dr. Miyazono (The Institute of Japanese Foundation for Cancer Research, Tokyo). The C2C12 cells were trypsinized and seeded in triplicate wells at 50,000 cells/well in 12-well plates on day 1. On day 2, the cells were co-transfected with the 9 × GCCG-reporter construct and the renilla-luciferase control vector using SuperFect (QIAGEN) for 24 hr. A total of 1 µg of plasmids was used for co-transfection in each well and the concentration of renilla-luciferase vector was 1/15 of the 9 × GCCG-reporter plasmid. On day 3, medium was replaced with DMEM containing 2% FBS and the cells were treated with various concentrations of compound. On day 4, the cells were treated with BMP-2. On day 5, the luciferase activities were measured as described above.

## RNA extraction and reverse transcription

The MC3T3 cells were plated at a density of 200,000 cells/well in 6-well plates and grown overnight in DMEM containing 10% FBS. On day 2, the culture medium was replaced with DMEM containing 2% FBS and the cells were treated with various concentrations of selected compound (diluted from 10 mg/ml stock solutions prepared in DMSO) for 24 hr. In control cultures, a DMSO solvent concentration of 0.01% (v/v) was applied. On day 3, medium was replaced with fresh DMEM containing 2% FBS and the cells were treated with recombinant human (rh)BMP-2 for 24 hr. Total RNA was harvested using the RNeasy Mini Kit according to the manufacturer's instructions (QIAGEN). The harvested RNA was digested with RNase-free DNase I (QIAGEN) to remove DNA contamination. The concentration of the isolated RNA was determined by measuring the absorbance at 260 nm wavelength with a spectrophotometer (Epoch, Biotek). The ratio of A260/A280 was between 1.9 and 2.0. Reverse transcription was carried out to synthesize cDNA in a 50 µl volume with 2 µg of total RNA, using the Invitrogen SuperScript IV VILO (Cat# 11756050) for 10 min at 25°C, 10 min at 50°C, and 5 min at 85°C.

## Quantitative real-time RT-PCR

Expression of murine β-catenin, AXIN2, BMP-2, ID1, RANKL, and OPG was analyzed in duplicates (primers given in *Supplementary file 1*). Gene transcription levels were determined with the comparative delta Ct method using 18S as a reference. Quantitative real-time RT-PCR was performed also to determine the mRNA expression level of early marker genes of BMP pathway. All primers were preverified and procured from Bio-Rad. Real-time PCR was performed with the following three-step protocol: step 1, 50°C for 2 min; step 2, 95°C for 10 min; step 3, 40 cycles of 95°C for 15 s and 62°C for 1 min using the 7500 real-time PCR System (Applied Biosystems, Foster City, CA). To confirm the amplification specificity, the PCR products were subjected to a dissociation curve analysis.

## SDS-PAGE and western blotting

We investigated the intensity and duration of the Smad1 signal by evaluating the levels of C-terminal phosphorylation (pSmad1$^{Cter}$) at different time points using murine MSCs as well as MC3T3-E1 cells, which are known to respond particularly well to BMP in serum-free medium. To start, recombinant human (rh)BMP-2 (50 ng/ml) was added for 15 min only. Next, pSmad1$^{Cter}$ was assessed via western blot at time 0 followed by 1, 2, and 3 hr thereafter. This was repeated with the SMI candidates (0–20 µM). SB415286 (40 µM), a known inhibitor of GSK3b, was also be used as a positive control. Similarly, we determined the levels of both phosphorylated-β-catenin and unphosphorylated-β-catenin using Wnt1-conditioned media, both in the presence and absence of recombinant murine sclerostin (rmSCL, 80 ng/ml), with and without the sclerostin SMIs (0, 2.5, 5, 7.5, 10, 15, and 20 µM) using murine MSCs as well as MC3T3-E1 cells as model systems.

Cells were lysed to obtain total protein using Mammalian Protein Extraction Reagent (Pierce Biotechnology, Rockford, IL) or lysed to obtain nuclear protein using NE-PER Nuclear and Cytoplasmic Extraction Reagents (Pierce Biotechnology) according to the manufacturer's protocol. Each sample (10 µg of protein) was mixed with NuPage loading buffer (Invitrogen, Carlsbad, CA) for a total volume of 20 µl and boiled for 5 min. The proteins were separated by electrophoresis under denaturing conditions on NuPage Bis-Tris Pre-Cast gels (Invitrogen) for 60 min at 200 V and transferred onto nitrocellulose membranes (Invitrogen) for 60 min at 30 V. After the transfer, the membranes were incubated in 25 ml of blocking buffer (5% non-fat dry milk in Tris buffered saline [TBS]) for 1 hr at room temperature. After blocking, membranes were washed three times for 5 min each in 15 ml of TBS with 0.1% Tween-20 (TBST). Washed membranes were incubated with different primary antibodies in TBST overnight at 4°C. Anti-actin antibody was purchased from Santa Cruz Biotechnology (Santa Cruz, CA); other antibodies were purchased from Cell Signaling Technology (Beverly, MA). After incubation with primary antibody, membranes were washed three times for 5 min each with 15 ml of TBST. Washed membranes were incubated with HRP-conjugated anti-rabbit or anti-mouse secondary antibodies as indicated (1:2000, Cell Signaling Technology, Beverly, MA) in 10 ml of blocking buffer with gentle agitation for 1 hr at room temperature. After incubation with secondary antibodies, membranes were washed three times for 5 min each with 15 ml of TBST. Washed membranes were incubated with 5 ml of SuperSignal West Pico Western blot substrate (Pierce Biotechnology) with gentle agitation for 4 min at room temperature. Membranes were drained of excess developing solution, wrapped in plastic wrap, and exposed to X-ray films.

## ALP assay

The C2C12 cells were plated at 200,000 cells/well in 6-well plates and grown overnight in DMEM containing 10% FBS. On day 2, the culture medium was replaced with DMEM containing 2% FBS and the cells were treated with 0.5 uM or indicated concentration of compound for 24 hr in 2 ml culture medium. On day 3, the cells were treated with a final concentration of 50 ng/ml of BMP-2 with or without compound in DMEM medium containing 2% FBS for 72 hr. The cells were washed with phosphate-buffered saline (PBS) and lysed by the addition of lysis buffer (10 mM Tris-HCl pH 8.0, 1 mM MgCl$_2$, and 0.5% Triton X-100). The cell lysates were centrifuged for 5 min at 13,000 × g. The supernatant was removed, and the aliquots were assayed for ALP activity and protein amount. The ALP activity was measured in triplicate using an ALP assay kit (Sigma-Aldrich, St. Louis, MO) in microtiter plates. The protein amount was determined with Bio-Rad protein assay reagent (Bio-Rad, Hercules, CA) using bovine serum albumin (BSA) as a standard. The ALP activity (nmoles of p-nitrophenol per ml) was normalized to the protein amount (nmoles of p-nitrophenol per µg).

## Mineralization assay

To quantify the calcium deposition (mineralization) of osteogenic samples, murine MSCs and MC3T3 cells were harvested on day 14. After washing the samples twice with PBS and disrupting the monolayer with a cell scraper, calcium ions were dissolved from the extracellular matrix by shaking in 500 µl/well 0.5 N HCl at 4°C for 4 hr. Calcium content was determined in technical triplicates with the QuantiChrom calcium assay kit (BioAssay Systems, Hayward, CA) according to the manufacturer's protocol. Protein content was measured in technical triplicates using Roti-Quant (Carl Roth, Karlsruhe, Germany) according to the manufacturer's instructions. Calcium concentrations were normalized to the protein content.

## Bone resorption assay

RAW 264.7 cells were plated at a low density in the presence of DMEM with 10% FBS. After 24 hr, the media was changed to α-MEM containing macrophage colony-stimulating factor (M-CSF; 10 ng/ml) and 66 ng/ml receptor activator of nuclear factor kappa-B ligand (RANKL, Lonza Biosciences). OC activity was measured by releasing the europium conjugated human type I collagen-coated on the bottom of the OsteoLyse Assay Kit (Lonza Biosciences) at each time point. Various treatments were performed as indicated in figure legends. Then, 200 µl of a fluorophore releasing reagent (Lonza Biosciences) was placed in each well of a 96-well black, clear-bottom assay plate (Corning Inc, Corning, NY). Also, 10 µl of cell culture supernatant was transferred to each well of the assay plate containing the fluorophore releasing reagent. The fluorescence of each well of the assay plate was measured with an excitation wavelength of 340 nm and an emission wavelength of 615 nm over a 400 µs period after an initial delay of 400 µs.

## Histological analysis of rat explants

After euthanasia, the subcutaneous implants were fixed with 10% formalin. Following fixation, the implants were washed and placed into a processor that dehydrated the samples in 70% alcohol, followed by 95%, 100%, and xylene. The samples were then embedded in paraffin and cut into slices of 5 microns using a microtome (Accu-Cut SRM 200 Rotary Microtomoe, Sakura Finetek USA, CA). Slides were stained with hematoxylin and eosin (H&E) and Goldner's trichrome (Sigma-Aldrich). Images were obtained with Lionheart LX (Biotek Instruments Inc, Winooski, VT) at 4× and captured using Gen 5 software.

## Rat subcutaneous ectopic model

Male 5- to 6-week-old Sprague–Dawley rats (Harlan Laboratories, Indianapolis, IN) were anesthetized with 1–2% isoflurane mixed with oxygen at a flow rate of 0.5–1 l/min and maintained during surgery with this same dose. Surgery was performed with the animal positioned supine on a circulating-water heating pad. Four 1-cm transverse incisions were made about 3 cm apart on the chest of each rat, and subcutaneous pockets were created by blunt dissection with scissors. The implants were inserted into the pockets and closure was accomplished with closely spaced interrupted absorbable polyglactin-910 sutures (Vicryl; Ethicon, Johnson & Johnson, Somerville, NJ). Each SMI was tested for its ability to produce de novo ectopic mineralization as a standalone factor at the following concentrations: 0,

10, 25, 50, and 100 mM. These were compared to the results obtained with two doses of rhBMP-2 that are known to produce consistent ectopic mineralization in this model (5 and 10 µg). Either SMI or BMP was loaded alone onto a plane collagen disc (8 mm diameter × 3 mm thickness). In each individual rat, four implants with different doses were implanted through separate skin incisions on the chest into subcutaneous pockets that did not communicate.

The rats were housed in autoclaved cages (two per cage) and were fed food and water ad libitum without restrictions on activity. All rats fed well after surgery. There were no postoperative complications associated with the surgical procedure. The rats were euthanized 4 wk postoperatively by $CO_2$ inhalation. The implants were harvested and evaluated by high-resolution digital radiography, micro-computed tomography (µCT), and non-decalcified histological analysis.

## Rabbit spine fusions

Thirty-six skeletally mature female New Zealand White rabbits (*Oryctolagus cuniculus*) weighing 3.1–3.7 kg were obtained from Covance (Princeton, NJ). All rabbits underwent single-level, bilateral, posterolateral intertransverse process fusions at L5–L6, exactly as described by *Boden et al., 1976*. In short, a dorsal midline skin incision was made in the lumbar region extending from L4–S1 using bony landmarks, followed by two paramedian fascial incisions. The intermuscular plane between the multifidus and longissimus muscles was developed to expose the transverse processes of L5 and L6 as well as the intertransverse membrane. In the rabbits in which ICBG was to be harvested, an additional fascial incision was made overlying each iliac crest to harvest ~1.5–2.0 ml corticocancellous autologous bone graft. The exposed TPs were then decorticated with an electric burr. The SMIs (either 300 or 500 mM) were then loaded onto a plain collagen sponge. These sponges ±ICBG were then placed between the decorticated TPs in the paraspinal bed. Fascial incisions were approximated using 3–0 absorbable suture and the skin was closed with staples. Ceftiofur sodium (Naxcel), 5 mg/kg, was injected subcutaneously as a preoperative antibiotic and an epidural block with morphine (0.01 µg/kg) was used in rabbits who had ICBG harvested. In all rabbits, a transdermal fentanyl patch (25 µg/hr) was for postoperative pain control. All rabbits were allowed to eat and perform activities ad lib. Animals were monitored closely and treated for pain in compliance with institutional guidelines. In compliance with current Animal Welfare Assurance standards, IV pentobarbital was used to euthanize rabbits 6 wk following arthrodesis.

## Radiography

X-ray scanning (In-Vivo Xtreme, Bruker Corp., Billerica, MA) was performed on the subcutaneous implants after formalin fixation. The scans were executed with an exposure time of 1.2 s and a voltage of 45 kV. The area of bone formation was determined based on the percent volume that an explant was mineralized on X-ray images (0 = no bone; 1 = <25% of the implant is mineralized; 2 = 25–49% of the implant is mineralized; 3 = 50–74% of the implant is mineralized; 4 = 75–99% of the implant is mineralized; 5 = 100% or greater of the original implant size is mineralized).

## Computed tomography

To further assess bone (rabbit spine fusions) or ectopic mineralization (rat SQ) formation, uCT scans (Micro-CT40, Scanco Medical, Bruttisellen, Switzerland) were performed. Samples were scanned with a 30 um voxel size at a voltage of 45 kVp and a current of 177 uA. Scanned images were reconstructed in the sagittal and coronal plane, and the formation of newly calcified tissue was evaluated quantitatively for volume and density.

## Histological sections of rabbit spine fusions

After tissue fixation, the fusion masses were decalcified and paraffin embedded. The serial sagittal sections were then deparaffinized, dehydrated in xylene, and serially incubated in an alcohol gradient (100%, 95%, 75%, 50% ethanol), followed by rehydration in distilled water. The samples were then stained with H&E. After staining, the sections were dehydrated by being serially incubated in an alcohol gradient of increasing concentration (50%, 75%, 95%, 100% ethanol), followed by incubation in xylene for 5 min. A drop of mounting medium was then added to each section, followed by placement of a coverslip. Sections were then imaged and merged at 5× magnification using a Leica DM6

B upright microscope to get representative images of the whole fusion masses (Leica Microsystems, Wetzlar, Germany).

## Statistics and calculations

All in vitro results are presented as the mean of three determinations (*n*), with error bars representing the standard error of the mean (SEM). Experimental results that are visually represented are from consistent experiments where one representative experimental result is shown. Statistical significance (p<0.05) was calculated using a one-way ANOVA with Bonferroni post hoc test (equal variances assumed) or Dunnett's T3 post hoc test (equal variances not assumed) using Statistical Products for Social Sciences Version 16.0 (SPSS 16.0) for Windows (SPSS, Chicago, IL) to compare various treatments in multigroup analysis. Statistical probability of p<0.05 was considered significant.

The sample size needed for statistical significance for the rat and rabbit experiments was determined by a priori power analyses. For the rat experiments, all doses of each SMI were performed in eight different animals to ensure sufficient power to detect a 20% difference in the amount of ectopic mineralization produced (p<0.05). For the rabbit spinal fusions, based on extensive experience with the model and considering an ~15% perioperative mortality rate, three rabbits per group, with two sides of the spine being assessed per animal (six total fusions per group), were used to achieve a statistical power to detect a 25% difference in fusion success (p<0.05).

## Study approval

All rat and rabbit surgeries and procedures were first approved (#VOOS-14 - 2016-020211) by the Atlanta VA Medical Center Institutional Animal Care and Use Committee (IACUC).

## Results

### Computational design and screening of anti-sclerostin small molecules

Receptor-based virtual screening of chemical libraries against a given target for the purpose of predicting the conformation and binding affinity of small molecules is becoming a popular practice in modern drug discovery (*Mahasenan and Li, 2012*). For our study, AutoDock Vina (*Trott and Olson, 2010*) and FRED (*McGann, 2011*) were used to find the effective inhibitors from a library of compounds against the loop 2 region of sclerostin, from amino acids at the 91st to 95th position, as well as the cystine knot region (i.e. Cys-85 and Cys-143) (*Trott and Olson, 2010*; *McGann et al., 2003*). Each compound was docked against both target regions of interest on sclerostin. The predicted binding energy from the dockings provided a ranking of compounds based on their binding affinities. We found that some ligands showed very high binding affinity to the Ile-95 residue of sclerostin (*Figure 1*). There were also other ligands that had high binding affinity for Cys-85 and Cys-143. The ligands with the highest binding affinity obtained from the docking programs were further analyzed for their toxicity using Toxtree v.2.6.0 (*Patlewicz et al., 2008*). Most of the high-affinity ligands obtained from FRED belonged to class II (an intermediate order of oral toxicity), indicating that they might be useful as clinical inhibitors of sclerostin. Candidate compounds were then selected in silico against the loop 2 region and the cystine knot region of sclerostin. We estimated the translational, rotational, and vibrational entropies of protein–drug complexes. Binding energy values were found to be low for several of the Ludi high-scoring molecules (*Figure 1C*).

The analysis of ligands showed that some of the SMI candidates bound strongly to multiple active sites (i.e. Cys-85 and Ile-95) on sclerostin. As such, these ligands were considered strong candidates because of the strong predicted hydrogen bonding with both active sites, as well as their bond distances of less than 2 Angstroms. Upon further evaluation of these unique candidates, we noted similarity in their structures as well, with most of them containing a heterocyclic ring with complex substituents.

### In vitro screening of lead sclerostin SMI candidates

From the in silico studies, we selected the 100 most promising compounds (defined as having a Ludi score >300 and class I or II toxicity) for in vitro screening in a biochemical assay that we have previously characterized and validated for this purpose (*Figure 1—figure supplement 1*; *Okada et al., 2009*). Specifically, each sclerostin SMI candidate was tested for its ability to enhance canonical Wnt signaling

**Table 1.** Based off cell-based screening assays, these two FDA-approved compounds were determined to be good candidates to be repurposed as a sclerostin small-molecule inhibitor (SMI) for bone regeneration.

MOA, mechanism of action.

| Drug name | MOA | Current use | Dosing | Metabolism |
|---|---|---|---|---|
| Fluticasone (F) | Glucocorticoid receptor agonist | Topical anti-inflammatory | 100–2000 µg/day | Hepatic |
| Valproic acid (VA1) | Unknown | Anti-epileptic drug, migraines | 15–60 mg/kg/day | Hepatic |

in vitro using a Wnt-specific TCF/LEF-driven Cignal reporter system (QIAGEN) that has been optimized for Wnt3a response. This reporter system uses a mouse myoblast cell line (C2C12), which have been stimulated toward the osteoblastic phenotype (*Okada et al., 2009*). This cell-based screening assay identified multiple sclerostin SMI candidates with the ability to enhance Wnt/β-catenin signaling (*Figure 1D*), as predicted in silico. For the FDA-approved compounds, we next performed a comprehensive literature review (data not shown) of the top six candidates from the in vitro screening, paying particular attention to what is currently known in the literature about each drug's mechanism of action, toxicity, dosing, side effects, and any specific effects on bone biology. Three candidate compounds were chosen for analysis, one novel drug (hereafter called C07) and two FDA-approved drugs (valproic acid and fluticasone, hereafter called VA1 and F, respectively) that have the potential to be repurposed as anti-sclerostin SMIs (*Table 1*).

## Sclerostin SMIs block binding of sclerostin to LRP5

To confirm that our top three sclerostin SMI candidates inhibit sclerostin's interaction with its receptor as predicted in silico, we performed an in vitro binding assay with purified recombinant sclerostin and LRP5 proteins. Sclerostin and LRP5 protein were labeled with biotin and $^{125}$Iodine, respectively. Constant amounts of $^{125}$Iodine-labeled LRP5 and biotin-labeled sclerostin were then incubated with or without varying concentrations of unlabeled-sclerostin with/without various concentrations of sclerostin SMIs. As expected, unlabeled sclerostin was found to compete off ~90% of the labeled sclerostin (*Figure 2A*). It was found that the interaction between sclerostin and LRP5 is saturable, competable, and concentration-dependent in our binding assay. All three sclerostin SMIs competed with sclerostin to prevent labeled sclerostin from binding to LRP5, confirming that our lead compounds significantly disrupt sclerostin binding to LRP5. The SMI C07 was the most effective, preventing ~40% of sclerostin from binding to LRP5.

## Sclerostin SMIs enhance canonical Wnt signaling

All three leading sclerostin SMI candidates were then tested for their ability to induce canonical Wnt signaling in vitro using a gene reporter assay system. To do this, a TCF/LEF-responsive reporter was transfected into both murine BMSCs and pre-osteoblast (MC3T3-E1) cells. In the presence of a suboptimal dose of 10 ng/ml of Wnt3a, all three SMIs were assessed for their ability to enhance Wnt/β-catenin signaling. By blocking endogenous sclerostin in these cells, F, VA1, and C07 were able to significantly increase canonical Wnt signaling intensity in a dose-dependent manner (*Figure 2B*). The efficacy of the individual SMIs were as follows: C07>VA1>F1. In addition, the presence of endogenous sclerostin protein was confirmed in both cell types via western blot to ensure that this observed effect was not due to an unexpected, off-target effect of the SMIs (*Figure 2—figure supplement 1*).

## Sclerostin SMIs increase the active form of intracellular β-catenin

Active canonical Wnt signaling results in intracellular accumulation of the active, non-phosphorylated form of β-catenin (*Baron and Rawadi, 2007*). As such, the ability of the SMIs to rescue the sclerostin inhibition of this process was also assessed. For this experiment, the cells were treated with or without sclerostin SMIs (10 µM) for 2 d. The cell lysates were subjected to western blotting with specific antibodies for phosphorylated and unphosphorylated forms of β-catenin. *Figure 2C* demonstrates that all three sclerostin SMIs can enhance canonical Wnt signaling by inhibiting the accumulation of the phosphorylated (inactive) form of β-catenin in both MC3T3-E1 and BMSC cells, supporting the notion

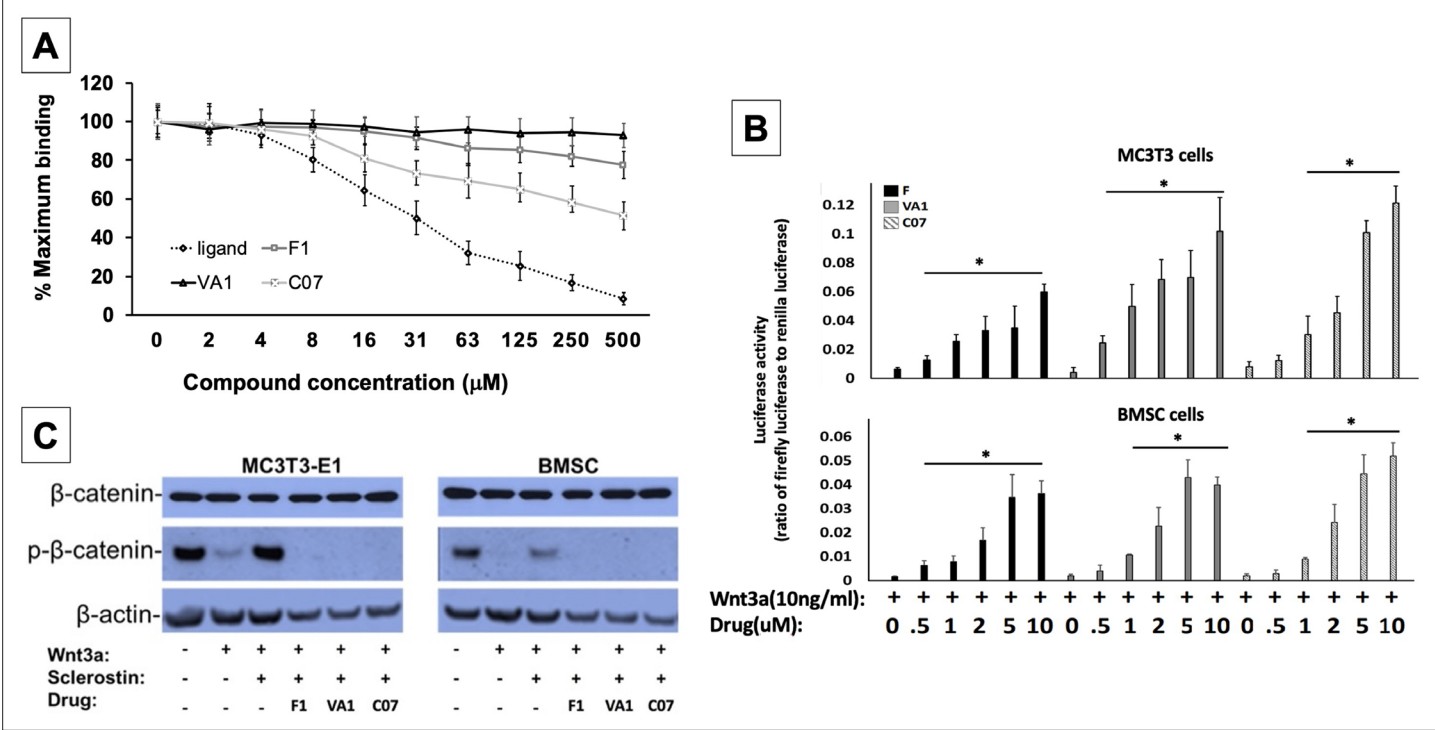

**Figure 2.** Sclerostin small-molecule inhibitors (SMIs) increase canonical Wnt signaling intensity in vitro. An in vitro binding assay with purified recombinant LRP5 and sclerostin proteins was used to assess for SMI–LRP interaction (**A**). The sclerostin SMIs were able to complete with sclerostin to prevent up to 40% of the labeled sclerostin from binding to LRP5, confirming that our lead compounds significantly disrupt sclerostin binding to LRP5. Moreover, the sclerostin SMIs were able to potentiate the intensity of Wnt signaling in vitro in a luciferase reporter assay system (**B**). A TCF/LEF-responsive reporter was transfected into murine MC3T3-E1 and MSC cells. In the presence of a suboptimal dose of 10 ng/ml of Wnt3a, all three SMIs dose-dependently enhanced Wnt/β-catenin signaling. Finally, sclerostin SMIs inhibit the accumulation of the phosphorylated (inactive) form of β-catenin in both murine MC3T3-E1 and MSC cells (**C**). Cells were treated with or without 10 μM concentration of sclerostin inhibitors for 2 d. Western blotting with antibodies specific for the phosphorylated and unphosphorylated forms of β-catenin was performed. All sclerostin SMIs inhibited the accumulation of phosphorylated (inactive) β-catenin in murine MC3T3-E1s and MSCs.

The online version of this article includes the following source data and figure supplement(s) for figure 2:

**Source data 1.** Raw β-catenin western blot data (*Figure 2*).

**Source data 2.** Raw sclerostin western blot data (*Figure 2—figure supplement 1*).

**Source data 3.** Relative luciferase activity of anti-sclerostin small molecules.

**Source data 4.** Binding assay raw data.

**Source data 5.** Raw data anti-sclerostin small-molecule candidate screening.

**Figure supplement 1.** Verification of endogenous sclerostin protein in cell types used.

that the increased activity observed in the Wnt gene reporter assay with sclerostin SMI treatment (*Figure 2B*) is likely due to direct enhancement of canonical Wnt signaling and β-catenin activity, rather than some unexpected off-target effect. In addition, this data suggests that the three sclerostin SMIs can effectively rescue osteoblasts from the inhibitory effects of endogenous and exogenous sclerostin in vitro.

## Sclerostin SMIs potentiate canonical BMP signaling

Each of the three sclerostin SMIs were next tested for their ability to potentiate the intensity of BMP signaling in vitro using an established BMP-responsive murine calvarial osteoblast reporter cell line (Kerafast Inc; *Yadav et al., 2012*). Since a suboptimal dose of BMP-2 is not established in this cell line, four doses of BMP-2 (0, 50, 100, and 200 ng/ml) were chosen to determine their potentiating effect on VA1, F1, and C07 (all at 10 μM concentration). Because the promoter was derived from early response genes for BMP, cells were first pretreated with compounds for 24 hr prior to treatment with BMP ±SMIs for 3 hr. As shown in *Figure 3A*, all three sclerostin SMI candidates exhibited a significant

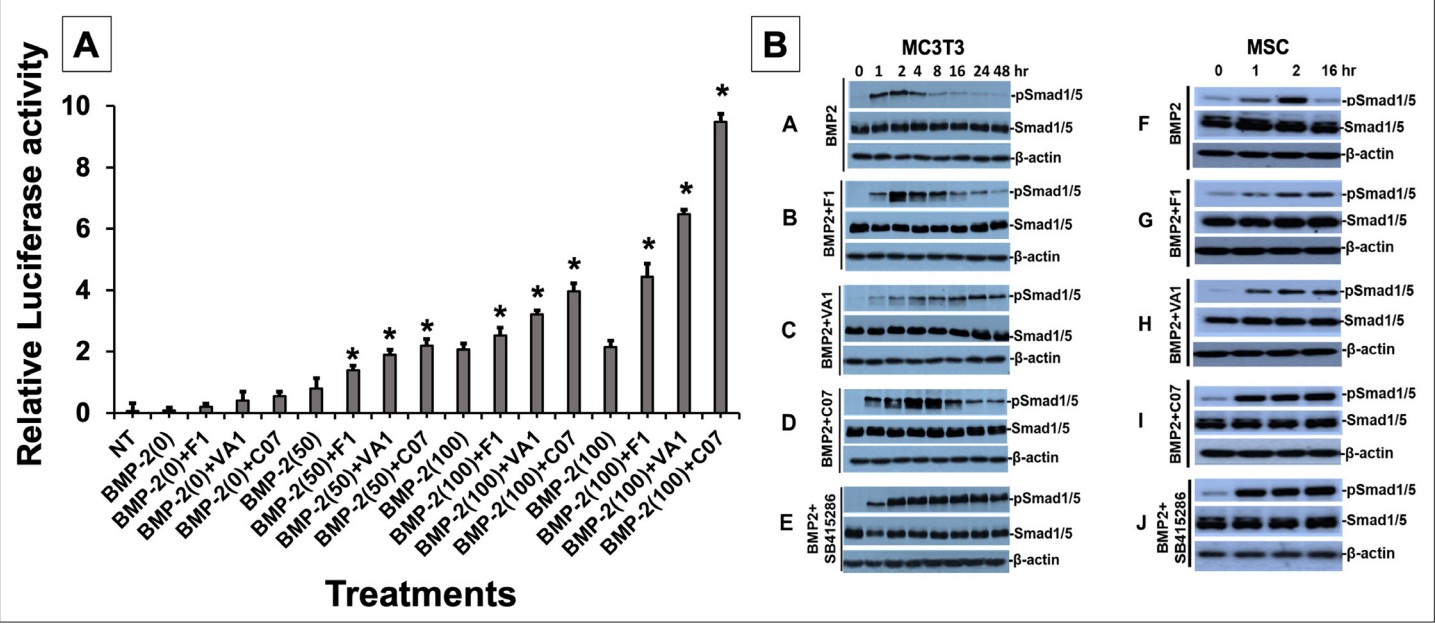

**Figure 3.** Sclerostin inhibitors potentiate the intensity and duration of BMP signaling in vitro. Using an established BMP-responsive murine calvarial osteoblast reporter cell line, enhancement of BMP-2-induced luciferase reporter activity by sclerostin small-molecule inhibitors (SMIs) was observed (**A**). Three doses (50, 100, and 200 ng/ml) of BMP-2 were chosen to determine potentiating effect of the SMIs at 10 μM concentration. Cells were pretreated with compounds for 24 hr followed by another treatment by BMP ± SMIs for 3 hr. Measurements were determined in triplicate (n = 3). In addition to potentiating BMP signal intensity, sclerostin SMIs were also able to increase accumulation of phosphorylated (active) Smads1/5 (**B**). Cells were treated with or without 10 uM concentration of sclerostin SMIs for 2 d. The cell lysates were then subjected to western blotting with specific antibodies for phosphorylated and unphosphorylated forms of Smads1/5. The GSK3b inhibitor SB415286 was a positive control.

The online version of this article includes the following source data for figure 3:

**Source data 1.** Pulse chase western blot raw data, MC3T3.

**Source data 2.** Pulse chase western blot raw data, MSC.

**Source data 3.** Relative luciferase activity of small molecule + BMP.

potentiating effect on BMP-induced luciferase activity at all doses of BMP-2 tested (50, 100, and 200 ng/ml). This enhancement demonstrated a dose-dependent manner. The efficacy of individual SMIs were as follows: C07>VA1>F1.

## Sclerostin SMIs increase BMP signaling duration

We next further explored other potential mechanisms for canonical Wnt/BMP crosstalk. While we already demonstrated an increased *intensity* in BMP response elements in the presence of our sclerostin SMIs (*Figure 3A*), we set out to assess whether the sclerostin SMIs also resulted in an increase in the *duration* of BMP signaling. We hypothesized that this increase in signaling duration may occur because the termination of intracellular BMP response to a given stimulus is typically determined by the activity of GSK3b, which itself is determined by phosphorylations that mark it for degradation in the proteasome. To test our hypothesis, we performed a BMP-2 pulse-chase experiment in which we assessed the levels of C-terminal phosphorylation of Smad1/5 (pSmad1/5$^{Cter}$) at different time points in murine BMSCs and MC3T3-E1 cells in the presence or absence of sclerostin SMIs. The GSK3b inhibitor SB415286 (*MacAulay et al., 2003*) was used as a positive control. As shown in *Figure 3B*, all three sclerostin SMIs significantly increased the duration of pSmad1/5$^{Cter}$ compared to untreated controls, with C07 resulting in the greatest increase in BMP signaling duration. These results suggest that after phosphorylation by the BMP receptor, the duration of the pSmad1/5$^{Cter}$ signal is controlled by canonical Wnt-mediated phosphorylation GSK3b. Taken together, out data suggests that the intensity of the Smad1/5/8 response is determined mostly by BMPs, but also canonical Wnt activation, while the duration of the Smad1/5/8 response is at least partially controlled by Wnt/β-catenin signaling via GSK3b phosphorylation.

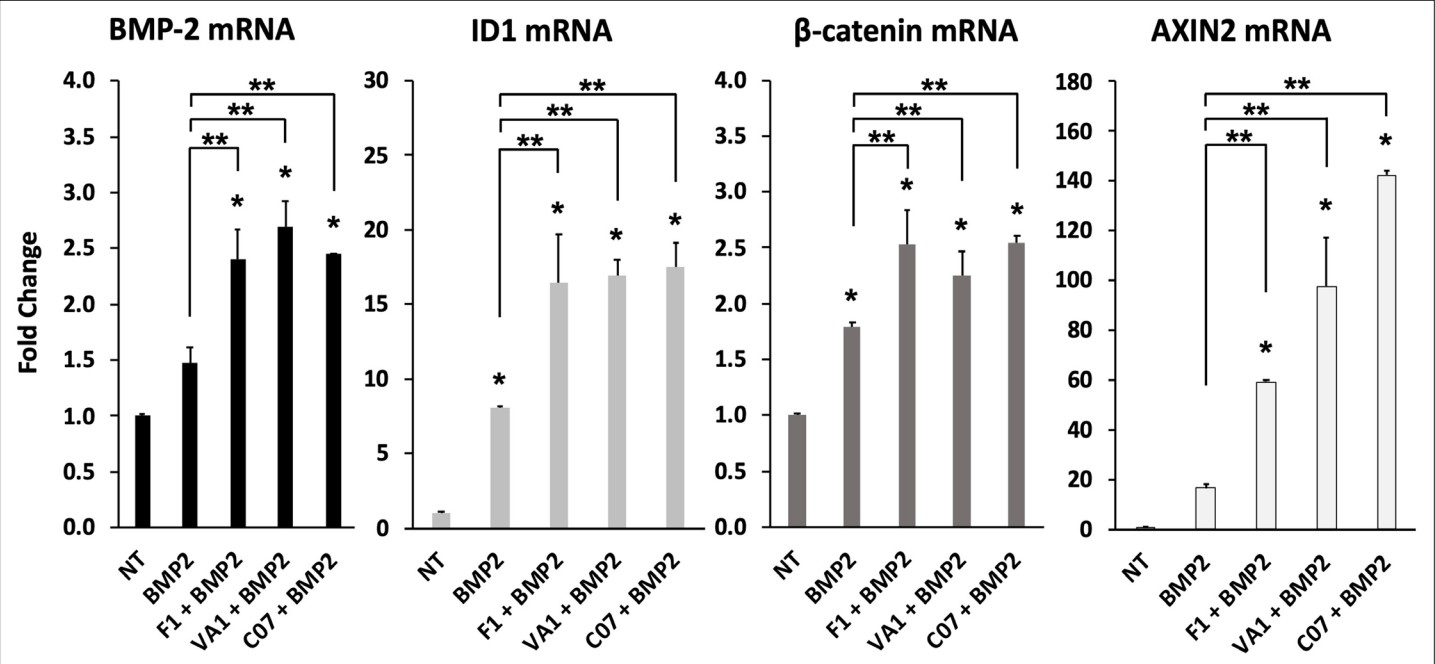

**Figure 4.** Sclerostin small-molecule inhibitors (SMIs) enhance the expression of osteogenic genes. The expression level of several genes was assessed by RT-qPCR after MC3T3 cells were treated with 10 µM of F1, VA1, or C07 in the presence of a suboptimal dose of BMP-2 (35 ng/ml). All data were determined in triplicate (n = 3). NT, no treatment (control). * indicates statistical significance (p<0.05) compared to NT, while ** indicates significance (p<0.05) from BMP-2 treatment alone.

The online version of this article includes the following source data for figure 4:

**Source data 1.** Raw PCR data.

## Gene expression with sclerostin SMI treatment

Next, the effect of the three sclerostin SMI candidates on the expression levels of specific genes that are important to canonical Wnt and BMP signaling were tested using RT-qPCR. These results are summarized in *Figure 4*. Both Axin2 (*Jho et al., 2002*) and Id1 (*Lewis and Prywes, 2013*) were chosen as markers because they are well known to be direct and specific target genes for the Wnt/β-catenin and BMP pathways, respectively (ID1 is relatively specific to BMPs and no other members of the TGF-β superfamily; *Korchynskyi and ten Dijke, 2002*). All three SMI candidates were able to significantly (p<0.05) increase the expression of Axin2 (60×, 100×, 140× over baseline for F1, VA1, and C07, respectively) and Id1 (16.5×, 17×, 17.5× over baseline for F1, VA1, and C07, respectively). All three SMIs were also able to significantly increase the expression of β-catenin over controls (p<0.05) as well (*Figure 4*, *). No change was seen in the expression of GSK3b (data not shown). In addition, all three SMI candidates were able to significantly increase the expression of endogenous BMP-2, while treatment with exogenous BMP-2 itself showed a small but insignificant increase (p>0.05). Moreover, when cells were treated with each of the sclerostin SMIs in addition to exogenous BMP-2, the expression of ID1, β-catenin, and Axin2 were all significantly greater compared to BMP-2 treatment alone (p<0.05) (*Figure 4*, **).

## Sclerostin SMIs enhance mineralization in vitro

Next, functional osteogenic activity was assessed in murine BMSCs and MC3T3-E1 cells upon treatment with the three sclerostin SMIs. Each SMI was tested for its ability to reverse the inhibitory effects of sclerostin on the osteogenic potential of these cells in an in vitro mineralization assay. When grown in osteogenic medium containing ascorbic acid (AA) and beta-glycerophosphate (BGP), these cells do not spontaneously differentiate and produce mineralization without a stimulus such as BMP-2. As shown in *Figure 5*, recombinant murine sclerostin (80 ng/ml) was able to completely reverse the osteoinductive effects of rhBMP-2 (20 ng/ml). This inhibitory effect was itself subsequently reversed by the addition of the three sclerostin SMI candidates at 10 µM. Thus, concurrent exposure to the sclerostin

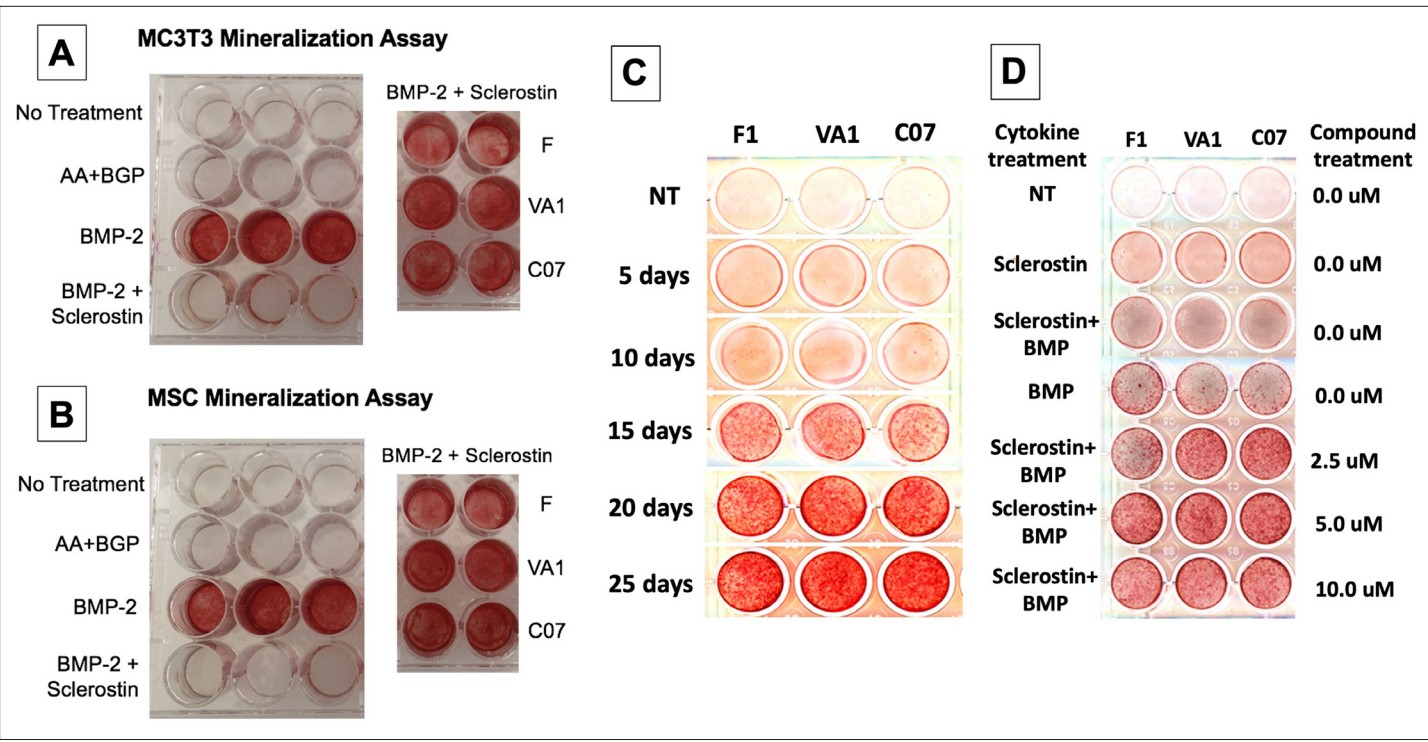

**Figure 5.** Sclerostin small-molecule inhibitors (SMIs) produce de novo mineralization in vitro. Sclerostin SMIs reverse the inhibitory effects of sclerostin on the osteogenic potential of murine pre-osteoblasts (MC3T3-E1 cells, **A**) and bone marrow stromal cells (BMSCs) (**B**) in a mineralization assay in vitro. Recombinant murine sclerostin (80 ng/ml) completely reversed the osteoinductive effects of rhBMP-2 (20 ng/ml). This inhibitory effect was subsequently reversed by the addition of the three sclerostin SMI candidates at 10 µM. Next, a time-course experiment was run (**C**), which shows that the SMIs accelerate mineralization over time. The dose–response on mineralization was also assessed, showing a ceiling effect around 10 µM (**D**).

SMIs effectively reversed the inhibitory effects of sclerostin to facilitate bone nodule formation in vitro in a dose-dependent and time-accelerated manner. This data confirms that these SMIs not only promote intracellular canonical Wnt and BMP signaling, but that those changes result in increased osteogenic activity.

## Sclerostin SMIs inhibit osteoclastic activity in vitro

We next investigated the effect of the sclerostin SMIs on gene expression within the RANKL-OPG axis since it has been previously found to be an example of crosstalk between the canonical Wnt and BMP pathways (*Kamiya et al., 2008*). As expected, the expression of RANKL in MC3T3-E1 cells increased more than 5-fold with BMP-2 treatment, while OPG only increased 1.8-fold (*Figure 6A*), leading to a >50% reduction in the OPG:RANKL ratio. This pro-osteoclastic stimulus, however, was then almost completely reversed by the addition of C07, but not F1 or VA1 ($p<0.05$), as evidenced by no change in RANKL expression compared to baseline but a significant increase in OPG expression (*Figure 6A*). These results were then confirmed at the protein level using OPG and RANKL western blot (*Figure 6B*).

Due to the ability of C07 to significantly increase the ratio of OPG:RANKL, we also investigated whether the sclerostin SMIs would be able to reduce the osteoclastic function of RAW 264.7 cells when induced by RANKL. RAW 264.7 cells were seeded onto an OsteoLyse (Lonza) plate at 10,000 cells/well and differentiated with soluble RANK ligand in the presence of various concentrations of ±Wnt3a; ±sclerostin; ±SMIs. At day 10 of culture, 10 µl samples of supernatant were removed and counted. As shown in *Figure 6C*, Wnt3a inhibits RANKL-induced osteoclast activity. The effect of Wnt3a could be reversed by treatment of cells with sclerostin. However, in the presence of our sclerostin SMIs, we observed the inhibition of osteoclast activity. The greatest functional effect was seen with C07. These exciting results suggest that the sclerostin SMIs can positively affect the functional activity of both osteoblasts and osteoclasts. Moreover, this data suggests that the sclerostin SMIs have a direct

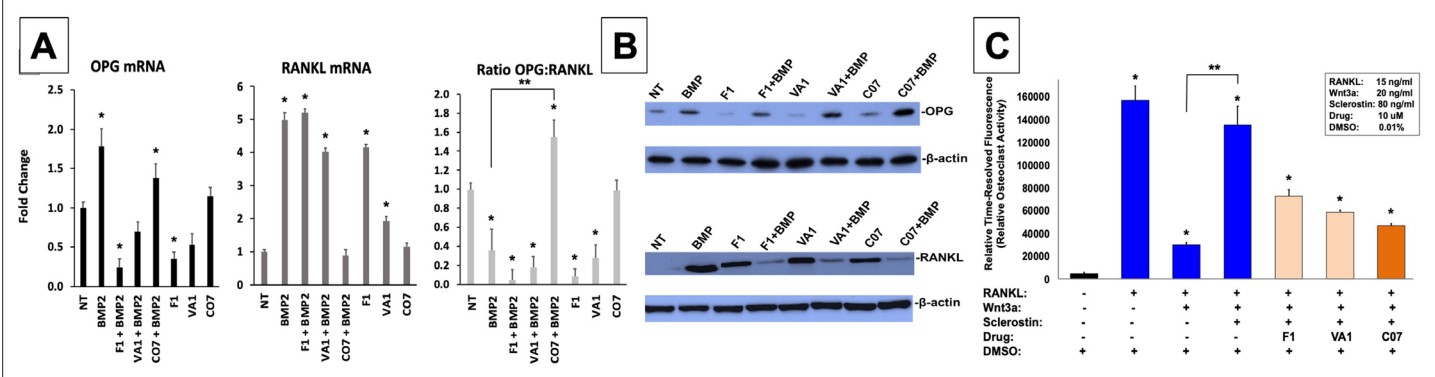

**Figure 6.** Sclerostin small-molecule inhibitors (SMIs) inhibit osteoclastic activity. The expression level of OPG and RANKL was assessed by RT-qPCR after MC3T3 cells were treated with 10 µM of F1, VA1, or C07 in the presence of a suboptimal dose of BMP-2 (35 ng/ml) (**A**). All data were determined in triplicate (n = 3). NT, no treatment (control). * indicates statistical significance (p<0.05) compared to NT, while ** indicates significance (p<0.05) from BMP-2 treatment alone. These changes in OPG and RANKL levels were also confirmed at the protein level using western blot, particularly in the presence of a suboptimal dose of BMP-2 (100 ng/ml) (**B**). Finally, all sclerostin SMIs were tested for their ability to inhibit osteoclast activity in RAW cells in a bone resorption assay. All three SMIs were found to significantly (p<0.05) inhibit sclerostin-induced osteoclast activity, with C07 approaching the response to Wnt3a+RANKL (**C**).

The online version of this article includes the following source data for figure 6:

**Source data 1.** Raw RANKL western blot data.

**Source data 2.** Raw OPG western blot data.

**Source data 3.** Raw PCR data.

**Source data 4.** Raw fluorescence data.

inhibitory effect on osteoclasts that is separate and in addition to its secondary effect on RANKL/OPG expression in MSCs or pre-osteoblasts.

## Standalone sclerostin SMIs produce de novo ectopic mineralization in vivo

Continuing to test for functional osteogenic activity, we next tested the two best-performing SMIs from the in vitro experiments (VA1 and C07) in a challenging in vivo rat subcutaneous ectopic mineralization model. Other than the osteoinductive BMPs, few if any proteins or small molecules are capable of inducing de novo ectopic mineralization in this model (*Akiyama et al., 2014*; *Minamide et al., 2003*; *Zanella et al., 2006*). Both SMIs were loaded individually as standalone agents onto a plain collagen sponge (DSM, Parsippany, NJ) at 0, 10, 25, 50, and 100 mM and then surgically implanted subcutaneously on the chest of 6-week-old male Sprague–Dawley rats for 4 wk. A positive control of 10 µg of recombinant BMP-2 was also tested. Plain radiographs and µCT data of the explants after 4 wk are shown in *Figure 7A and B*. Local subcutaneous delivery of both VA1 and C07 resulted in significant de novo ectopic mineralization as standalone agents, with both drugs showing a clear dose–response in this model. A 200 mM dose of both VA1 and C07 was also tested, which resulted in no discernible increase in mineralization over 100 mM (data not shown). As such, a 50 mM dose of VA1 (700 µg) and C07 (1.3 mg) was found to have a comparable osteogenic effect to 10 µg of BMP-2 in this model. The radiographic data of the explants was also confirmed by histology. Both H&E and Goldner's trichrome staining confirmed the presence of ectopic mineralization with VA1 and C07 treatment (*Figure 7C*). Also, neither C07 nor VA1 produced an adipogenic response, which was quite robust in the rhBMP-2 controls (*Figure 7C*).

Of note, this data is the first formal demonstration that locally delivered small-molecule drugs with the ability to inhibit the function of endogenous sclerostin can produce de novo ectopic mineralization in a non-bone environment as standalone osteoinductive agents in vivo.

## Sclerostin SMIs enhance spinal fusion rates in vivo

Next, both C07 and VA1 were assessed for their ability to enhance spinal fusion rates in vivo using a validated rabbit model of posterolateral lumbar fusions (*Boden et al., 1976*). Both SMIs were tested

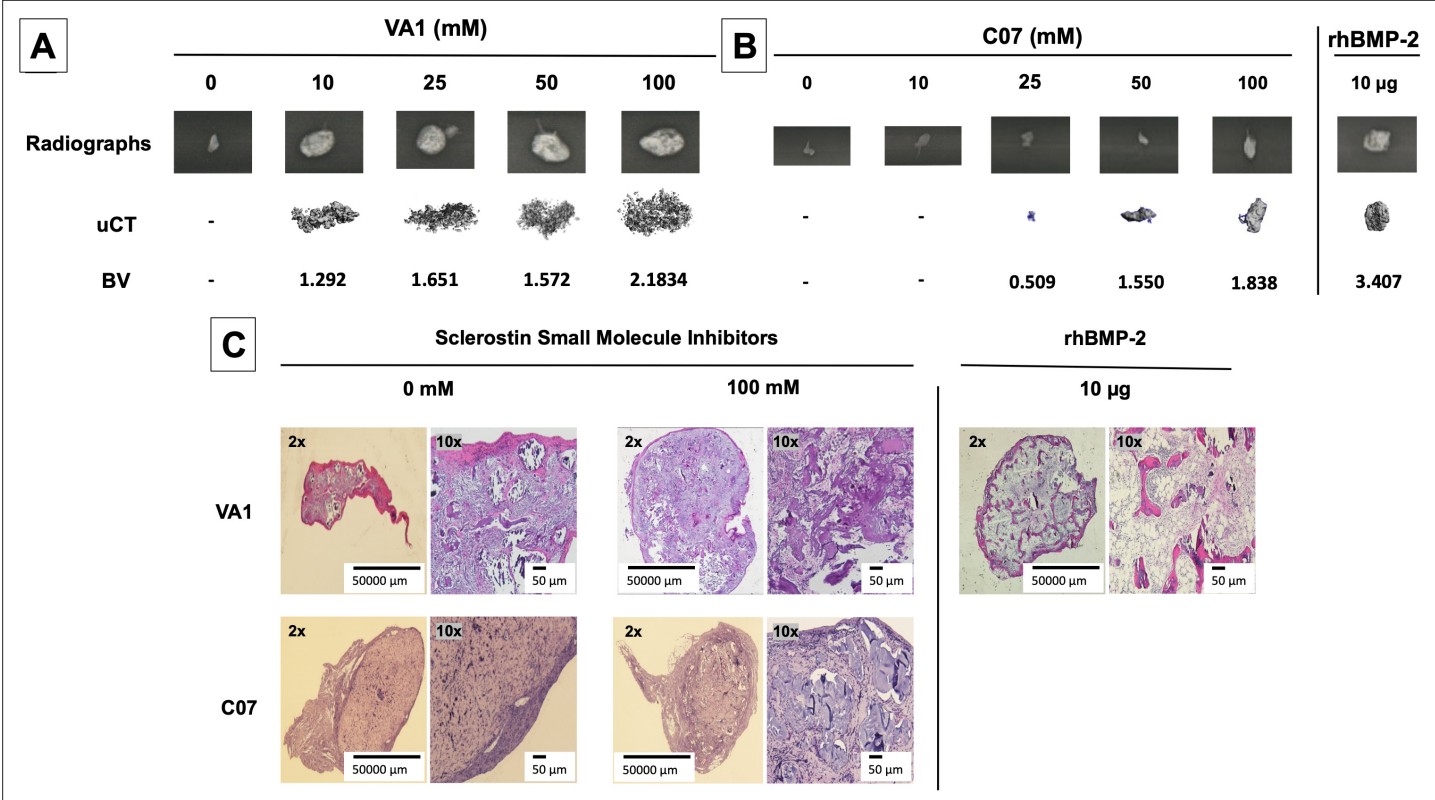

**Figure 7.** Standalone sclerostin small-molecule inhibitors (SMIs) result in de novo ectopic mineralization in vivo. A dose-dependent increase in ectopic mineralization was observed when VA1 and C07 were placed subcutaneously in rats on a plain collagen sponge (**A, B**). Significant differences compared with control implants treated with the vehicle alone (DMSO) were found at all doses tested. Plain radiographs (XR) and μCT, along with the corresponding BV (bone volume) values, are shown for both VA1 (**A**) and C07 (**B**). Of not, both VA1 and C07 were able induce ectopic mineralization without the addition of exogenous BMP. A 10 μg rhBMP-2-positive control is also shown (**B**). Of note, the scale of each μCT scan is consistent across all images. Representative histological images (hematoxylin and eosin) of subcutaneous explants are also shown (**C**), with 100 mM of VA1 and C07 demonstrating ectopic bone formation without the adipogenesis also seen with BMP-2.

as standalone osteoinductive drugs, as well as in combination with autologous ICBG, using two separate doses (300 and 500 mM). All rabbits were euthanized 6 wk following arthrodesis surgery and the spine fusion masses were assessed by both plain radiography and μCT. Successful fusion, defined as continuous bridging bone between the TPs, was assessed by two experienced spine surgeons (SBD and SMP). Both surgeons had to agree for a fusion to be considered successful.

When C07 was used at a dose of 500 mM in combination with ICBG, the posterolateral spine fusion rate was significantly increased compared to controls with ICBG alone (83% vs. 66%, p<0.05) (*Figure 8A*). Similarly, when the higher dose of VA1 (500 mM) was used alongside autologous ICBG, the fusion rate was also significantly increased compared to controls with ICBG alone (80% vs. 66%, p<0.05) (*Figure 8B*). When both C07 and VA1 were used at 500 mM as standalone drugs on a plain collagen sponge without ICBG, 33 and 17% of the spines successfully fused, respectively, which is significantly higher than the 0% fusion rate in this model when the transverse processes are decorticated alone (*Boden et al., 1976*). Neither VA1 nor C07 at 300 mM showed any increase in spinal fusion rates compared to controls.

Finally, because of the especially encouraging results of C07 to achieve posterolateral spine fusions as a standalone osteogenic drug, we next tested C07 at doses of 500 and 750 mM without ICBG (n = 6). As before, one dose was used on either side of the spine so that each rabbit acted as its own internal control. Then, 6 wk following arthrodesis surgery, it was found that the high dose of C07 (750 mM) produced a successful posterolateral spinal fusion in 80% of the rabbits (*Figure 8C*). Histological sections of these spinal fusion beds demonstrate membranous bone formation at the periphery of the fusion masses (emanating from each TP) as well as endochondral bone formation toward the center of the fusion masses (*Figure 9*), which is the expected pattern on bone formation

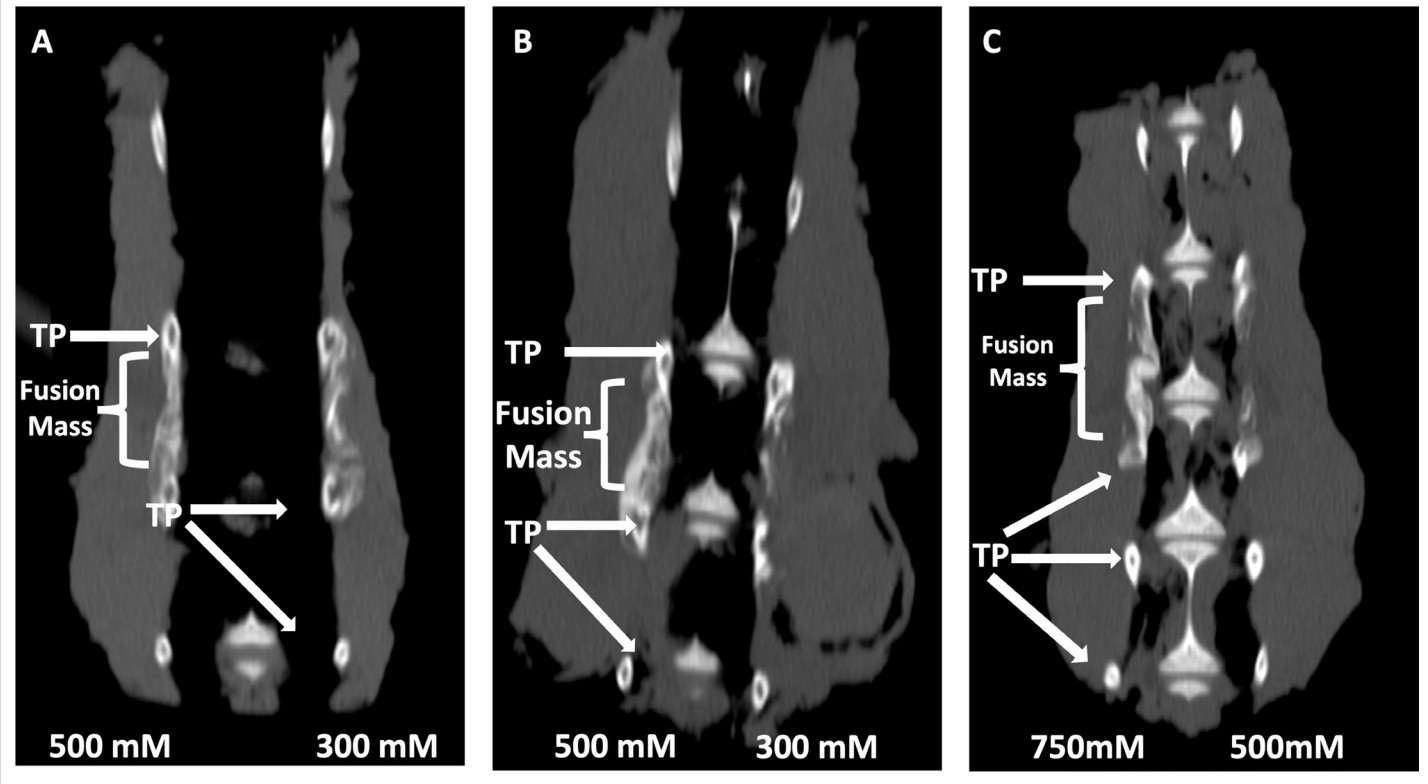

**Figure 8.** Locally delivered sclerostin small-molecule inhibitors (SMIs) produce successful spine fusions in vivo. Coronal μCT reconstructions of rabbit spines 6 wk following posterolateral spine arthrodesis are shown. In rabbits that received C07 along with autologous Iliac crest bone graft (IBCG) (**A**), the posterolateral spine fusion rate was significantly increased compared to controls with iliac crest bone graft (ICBG) alone (83% vs. 66%, p<0.05). Similarly, the fusion rate in rabbits that received VA1 with autologous ICBG (**B**) was also significantly increased versus ICBG control (80% vs. 66%, p<0.05). Lastly, standalone C07 (no ICBG) at a high dose of 750 mM (**C**) produced a fusion rate of 80% (vs. 0% in controls, p<0.05). A continuous bridge of bone from TP to TP (noted as 'Fusion Mass') is shown on the side with the higher dose in all instances (**A–C**).

in this model (*Schimandle and Boden, 1994*). Interestingly, no significant differences in bone quality are seen between those animals that received C07 alone (*Figure 9A*) and those that received C07 and autologous ICBG (*Figure 9B*).

Taken together, these data demonstrate a clear dose–response for C07 in this model. All spine fusion results for both SMIs are summarized in *Table 2* and representative X-rays are also shown in *Figure 9—figure supplement 1*.

### No evidence of hepatic toxicity in vivo

It is important to note that across the range of concentrations tested, none of the three sclerostin SMIs were found to be toxic to any of the cell types used in the in vitro experiments. The three SMIs were found to have no effect on murine MC3T3s, MSCs, or calvarial osteoblast cell number, morphology, or total protein yields while in culture for up to 20 d (data not shown). In addition, neither VA1 nor F1 caused changes in hepatic function labs 4 wk after SQ implantation in the rats at any of the doses tested, including a total dose per animal of 1000 mM in rats (*Supplementary file 2*). In addition, a total dose of C07 in rabbits of 2000 mM also showed no significant change in hepatic function 6 wk after implantation (*Supplementary file 2*).

### Discussion

Given the individual shortcomings of the bone graft options currently available for clinical use (*Boden, 2000*), in combination with the poor outcomes and increased healthcare costs associated with the development of a pseudarthrosis after fracture repair or spinal fusion (*Yeramaneni et al., 2016*; *Kim and Michelsen, 1992*), there is a clear clinical need for the development of additional biological

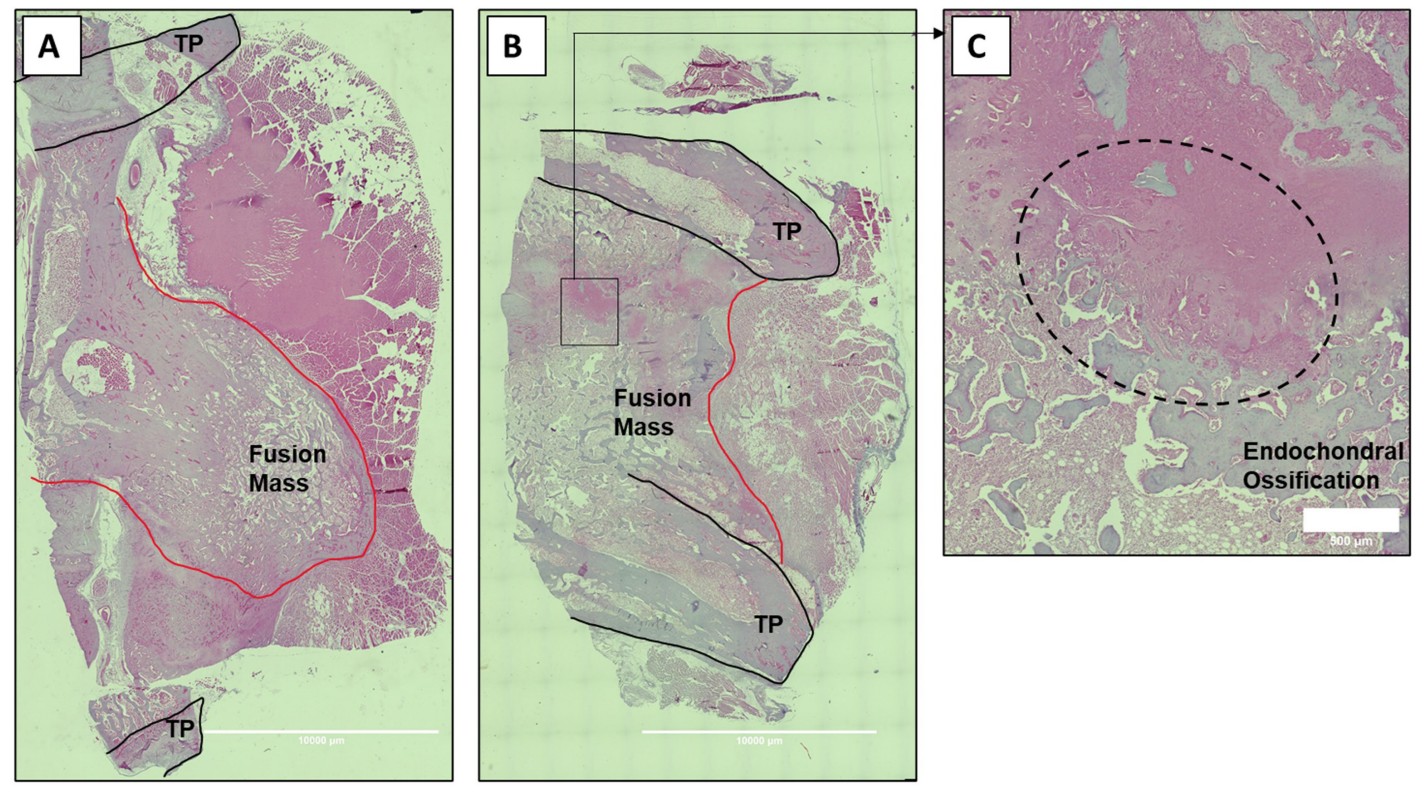

**Figure 9.** Histologic sections of C07-enhanced posterolateral spine fusions in vivo. Mid-coronal histological sections (5× magnification) from rabbit spinal fusion beds 6 wk following posterolateral spine arthrodesis were H&E-stained and confirm successful spinal fusion, showing bridging bone from TP to TP. Sections from a rabbit that received C07 without (**A**) and with (**B**) autologous IBCG are shown. Panel (**C**) shows a ×10 magnification of the boxed area in panel (**B**) and shows the area of endochondral ossification in the middle of the developing fusion mass.

The online version of this article includes the following figure supplement(s) for figure 9:

**Figure supplement 1.** Radiographs of rabbit posterolateral spine fusions.

options that surgeons can use at the time of surgery in order to consistently achieve bony union in these difficult clinical settings. Building on the pioneering work of systemic delivery of anti-sclerostin mAbs, we report a novel biological strategy to stimulate osteogenesis based on the local delivery of sclerostin SMIs. Herein, we have formally demonstrated that our sclerostin SMIs have the ability to enhance Wnt/β-catenin and canonical BMP signaling in vitro (in both BMSCs and pre-osteoblasts) and in vivo, resulting in increased osteogenesis while simultaneously decreasing bone resorption activity. This data is consistent with what has been previously been reported in multiple preclinical and clinical studies with the use of SOST gene knockout and anti-sclerostin mAbs, including (1) increased lineage commitment of BMSCs toward osteogenic cells (*Rawadi et al., 2003*; *Ross et al., 2000*), (2) increased osteoblast bone deposition (*Padhi et al., 2011*), and (3) simultaneous potentiation of BMP signaling intensity and duration,(*Eivers et al., 2009*; *Hussein et al., 2003*) and inhibition of osteoclast activity (*Delgado-Calle et al., 2017*; *McClung et al., 2014*).

The evolutionarily conserved canonical BMP and Wnt pathways are independent signaling mechanisms with different ligands, receptors, and intracellular signal transducers. However, these two pathways cooperatively control bone formation. Molecular studies have shown that functional communication between the canonical BMP and Wnt pathways involves multiple mechanisms. For example, BMP receptor 1A (BMPR1A) signaling directly upregulates the expression of sclerostin, leading to an inhibition of canonical Wnt signaling and a decrease in bone mass by the upregulation of osteoclastogenesis through the RANKL-OPG pathway (*Figure 10*; *Kamiya et al., 2008*). Interestingly, we observed a protective effect of C07 in the presence of a suboptimal dose of BMP-2 as C07 was able to completely reverse the significant increase in RANKL expression and protein levels caused by BMP-2

**Table 2.** Sclerostin small-molecule inhibitors (SMIs) increase posterolateral spinal fusion rates. The posterolateral spine fusion rate for each study is shown. The addition of either VA1 or C07 (500 mM) to iliac crest bone graft (ICBG) significantly increased the fusion rate compared to ICBG alone. Moreover, when compared to no treatment, the local delivery of either VA1 or C07 as a standalone osteoconductive drug resulted in significantly greater spine fusion rates in all conditions tested. *p<0.05.

|  | Fusion rate (%) |
| --- | --- |
| ICBG | 66 |
| VA1 + ICBG | 80 |
| C07 + ICBG | 83 |
| Control | 0 |
| VA1 | 17 |
| C07 | 33 |
| Control | 0 |
| VA1 | – |
| C07 | 80 |

The online version of this article includes the following source data for table 2:

**Source data 1.** Source data for *Table 2*.

treatment. In addition, we also observed that the sclerostin SMIs were able to inhibit bone resorption in a functional assay.

A second important way in which crosstalk exists between these two signaling pathways is that Wnt signaling directly regulates BMP/Smad1 signal termination (*Eivers et al., 2009*). Smad1, one of the major downstream signal transducers of the BMP pathway, contains GSK3b phosphorylation sites in its linker region (*Fuentealba et al., 2007*). As mentioned above, GSK3b is a critical factor in the canonical Wnt signaling pathway that, in the absence of canonical Wnts, plays a critical role in the phosphorylation and subsequent proteasomal degradation of β-catenin. It has recently been found that GSK3b phosphorylation is also required for the ubiquitination of Smad1 (*Fuentealba et al., 2007*). It is known that BMP signaling triggers sequential Smad1 phosphorylation by BMPR and GSK3b, followed by polyubiquitination as part of BMP's own negative feedback loop (*Eivers et al., 2009*). Once Smad1 is targeted for degradation, it is transported to the centrosome, where it is degraded by proteasomes. Interestingly, this process of Smad1 phosphorylation by GSK3b appears to be regulated directly by Wnt signaling, as demonstrated herein. We were also able to demonstrate that canonical Wnt signaling inhibits Smad1 phosphorylation by GSK3b and therefore stabilizes pSmad1$^{Cter}$, which is the active form of Smad1 that is phosphorylated at its C-terminal end by BMPR (*Eivers et al., 2009*). This suggests that the inhibition of GSK3b phosphorylation via canonical Wnt activation directly regulates the duration of the Smad1/5/8 signal (canonical BMP signaling) in a positive manner. Our data therefore confirm that sclerostin inhibition and canonical Wnt activation can simultaneously increase the signaling intensity and duration of the Smad1/5/8 response in osteoblasts (*Kamiya et al., 2008*).

Thirdly, and perhaps the most compelling mechanism highlighting the cooperation between the canonical BMP and Wnt pathways, is the transcriptional regulation of their common target genes, which harbor both Smad and TCF/Lef1 response elements. Smads can form a transcriptional complex with β-catenin/TCF/Lef1 and co-activate transcription of many target genes through these binding elements in response to both BMP and Wnt signaling (*Hussein et al., 2003*; *Labbé et al., 2000*). More recently, Zhang and colleagues demonstrated that the Wnt/β-catenin signaling pathway is a direct upstream activator of BMP-2 expression in MC3T3-E1 cells in vitro (*Zhang et al., 2013*). Our data confirmed this finding in multiple ways. In the presence of the three sclerostin SMIs tested, we found a significant (p<0.05) increase in BMP reporter activity and a significant (p<0.05) overexpression of BMP-2 and Id1. These results provide further insight into the nature of the functional crosstalk between these pathways with respect to skeletal homeostasis. Taken together, our data suggest that sclerostin inhibition can result in a multipronged push toward osteogenesis as it blocks the RANKL-mediated osteoclastogenic response to endogenous BMP-2 while simultaneously enhancing both canonical Wnt and BMP signaling.

Another noteworthy finding in this study is the location of the de novo mineralization seen in vivo using the sclerostin SMIs. Given that sclerostin is mostly produced by osteocytes, one could wonder how anti-sclerostin SMIs produce local mineralization in a subcutaneous environment. One possibility is that sclerostin is not solely produced by osteocytes and does not just effect bone; rather, other cell types are increasingly being found to be affected by sclerostin through the systemic circulation

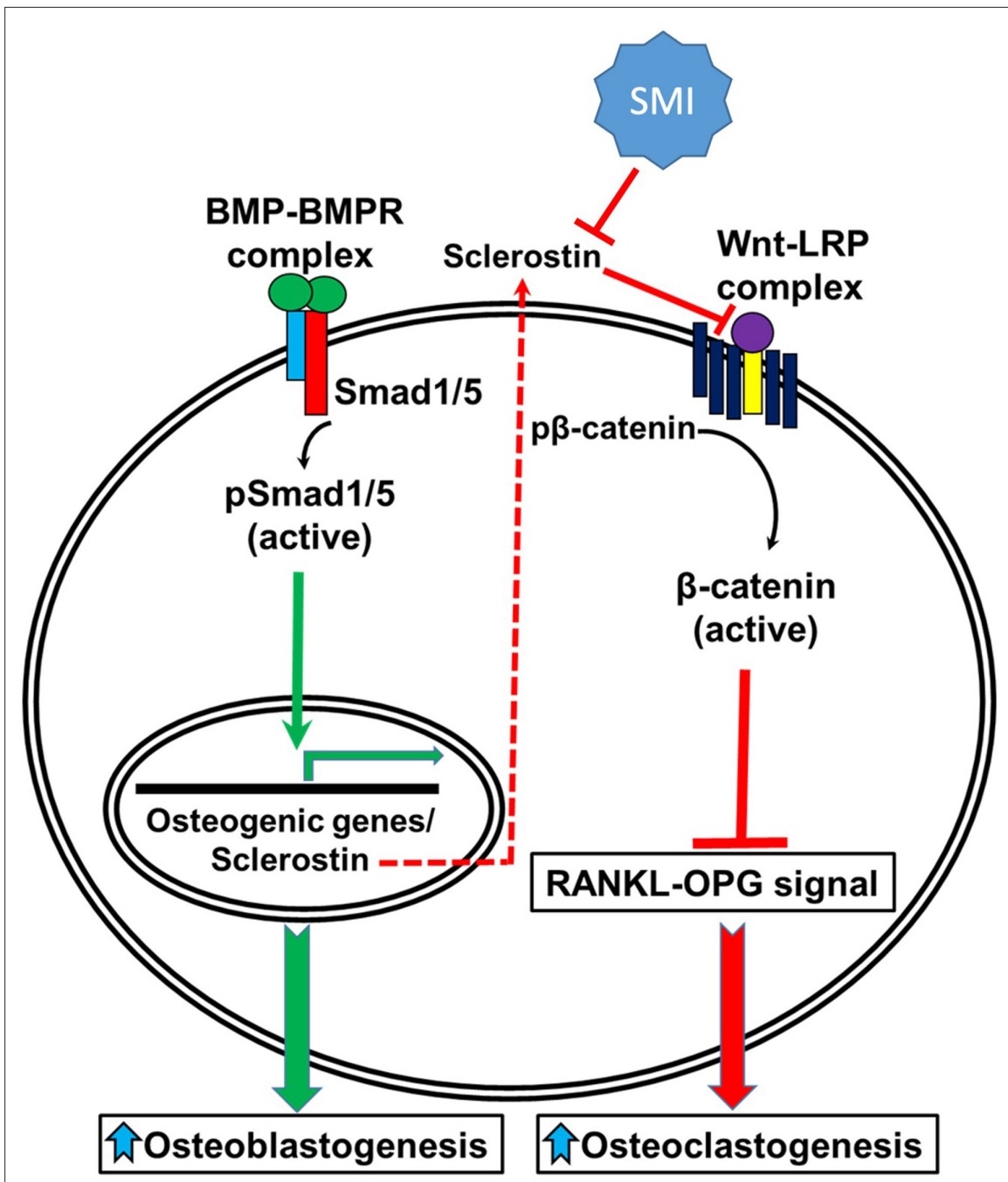

**Figure 10.** A schematic model of BMP-Wnt pathway crosstalk. Canonical BMP signaling results in upregulation of sclerostin expression (the sclerostin promoter has BMP response elements) as part of a negative osteogenic feedback mechanism. Inhibition of sclerostin, such as with a sclerostin small-molecule inhibitor (SMI), allows for reactivation of normal canonical Wnt signaling with subsequent downregulation of osteoclastogenesis (via inhibition of the RANKL-OPG pathway and direct inhibitory effects on osteoclasts) and further potentiation of canonical BMP signaling via GSK3b. As such, sclerostin is a key bone mass-regulating factor. GSK3b, glycogen synthase kinase 3 beta.

(*Battaglino et al., 2012*). For example, there is increasing evidence of sclerostin's effects on skeletal muscle, which is strengthened by the fact that LRP5/LRP6 are expressed in human muscle and that the Wnt/β-catenin signaling pathway is anabolic in muscle (*Tagliaferri et al., 2015*; *Karczewska-Kupczewska et al., 2016*; *Takeda et al., 2014*; *Rudnicki and Williams, 2015*). Interestingly, Huang and colleagues recently reported that Wnt3a, an osteocyte-derived Wnt-signaling agonist, promoted C2C12 cell differentiation in vitro and that sclerostin co-incubation (100 ng/ml) prevented this effect,

demonstrating that sclerostin negatively regulates Wnt-signaling in a mouse skeletal muscle cell line, at least when present in relatively high concentrations (*Huang et al., 2017*). Likewise, we show herein that sclerostin is also produced in C2C12 cells.

Although several small molecules that are capable of enhancing bone formation (and even BMP activity) have been described in the literature, few if any possess the osteoinductive potency of BMP itself – that is, the ability to form de novo ectopic mineralization as a standalone agent (*Wong et al., 2013*; *Montgomery et al., 2014*). Our observation that VA1 and C07 are able to leverage the complex crosstalk between canonical Wnt and BMP signaling on multiple levels and produce de novo ectopic mineralization in a non-bone environment as standalone osteoinductive agents in vivo makes them highly attractive as potential novel biologics for the induction of local bone healing in various clinical settings. This is particularly true for VA1 because it is an FDA-approved compound (for a different use) and already has an established safety profile in humans.

A strategy of locally delivered sclerostin SMIs has many important advantages for local bone formation over current existing systemically delivered mAb strategies. For example, non-local delivery strategies raise concerns over systemic exposure and the potential for off-target side effects. In addition, mAb-based strategies have other limitations for routine clinical use, including (1) limited mAb stability and shelf-life make storage of a commercially available product difficult, (2) increased difficulty with engineering mAbs into carriers or implants for local delivery, and (3) difficult production that is vulnerable to batch-to-batch variation and contamination. Monoclonal antibodies also have a very high manufacturing cost, which is not to be understated as we move toward a more value-based healthcare economy. As an alternative to mAbs, small-molecule drugs address each of these concerns. In addition, the relatively small size of SMIs compared to mAbs is thought to afford them immunoprivilege (*Blaich et al., 2010*). We believe that these factors significantly increase the attractiveness of using pharmacological sclerostin SMIs over existing mAb strategies.

Despite these encouraging early results, several limitations of this study merit mention. First, a significant limitation of this study is unpredictable translation of in vivo data from rats and rabbits to more clinically relevant models. However, we are actively exploring the use of these sclerostin SMIs in animal models of critically sized fracture defects as well. Another limitation of this study is that we did not definitively confirm that the SMIs were having their biological effect through their intended local mechanism and not via some unknown off-target effect. Future studies measuring systemically circulating sclerostin could answer this question. That being said, all of the in vitro data presented here confirm that all SMIs work via interrupting the interaction between sclerostin and its receptor, LRP5/6, thereby promoting canonical Wnt and BMP signal intensity and duration.

In conclusion, we identified several small molecules (VA1 and C07) that function by blocking the interaction of extracellular sclerostin and its receptor, LRP5/6. We show that these sclerostin SMIs strongly enhance both local canonical Wnt and BMP signaling and that they are able to drive osteogenesis in vitro and in vivo as standalone agents, which is a rare attribute for proteins or small molecules. We further show that this osteogenic effect is accomplished via enhancement of both Wnt/β-catenin signaling and canonical BMP signaling through a GSK3b-mediated effect on the duration of Smad1 phosphorylation. Our data indicate that these sclerostin SMIs may have potential as novel cost-effective biological bone graft enhancers or substitutes for either the repair of large critical-sized fracture defects and/or spinal fusions.

## Acknowledgements

This work was supported by a Veterans Affairs Career Development Award, Office of Research and Development Biomedical Laboratory Research & Development (IK2-BX003845) to SMP. We would also like to acknowledge Colleen Oliver and Mesfin Teklemariam for their help in performing the in vivo experiments. HD is supported by the VA Office of Research and Development Biomedical Laboratory Research & Development Service Awards (I01 BX004708 and I01 BX004878). GRB is also supported by the VA Office of Research and Development Biomedical Laboratory Research & Development Service Award (I01BX001516). We would also like to thank DSM (Exton, PA) for kindly donating some of the collagen sponges used in the rat ectopic mineralization experiments. The content of this manuscript is solely the responsibility of the authors and does not represent the views of the Department of Veterans Affairs or the United States Government.

## Additional information

### Funding

| Funder | Grant reference number | Author |
|---|---|---|
| U.S. Department of Veterans Affairs | IK2-BX003845 | Lorenzo M Fernandes |
| U.S. Department of Veterans Affairs | I01BX001516 | George R Beck |
| U.S. Department of Veterans Affairs | I01 BX004708 | Hicham Drissi |
| U.S. Department of Veterans Affairs | I01 BX004878 | Hicham Drissi |

The funders had no role in study design, data collection and interpretation, or the decision to submit the work for publication.

### Author contributions

Sreedhara Sangadala, Data curation, Investigation, Methodology, Writing - original draft, Writing – review and editing; Chi Heon Kim, Data curation, Investigation, Methodology, Writing – review and editing; Lorenzo M Fernandes, Data curation, Formal analysis, Supervision, Methodology, Writing – review and editing; Pooja Makkar, Resources, Data curation, Formal analysis, Supervision, Funding acquisition, Methodology, Writing – review and editing; George R Beck, Resources, Data curation, Formal analysis, Supervision, Funding acquisition, Investigation, Methodology, Writing - original draft, Project administration, Writing – review and editing; Scott D Boden, Resources, Supervision, Funding acquisition, Methodology, Writing – review and editing; Hicham Drissi, Resources, Data curation, Formal analysis, Funding acquisition, Methodology, Writing – review and editing; Steven M Presciutti, Conceptualization, Resources, Data curation, Software, Formal analysis, Supervision, Funding acquisition, Validation, Investigation, Visualization, Methodology, Writing - original draft, Project administration, Writing – review and editing

### Author ORCIDs

Hicham Drissi ⓘ http://orcid.org/0000-0002-3322-281X
Steven M Presciutti ⓘ http://orcid.org/0000-0001-6547-9495

### Ethics

This study was performed in strict accordance with the recommendations in the Guide for the Care and Use of Laboratory Animals of the National Institutes of Health. All rat and rabbit surgeries and procedures were first approved (#VOOS-14 - 2016-020211) by the Atlanta VA Medical Center Institutional Animal Care and Use Committee (IACUC).

### Decision letter and Author response

Decision letter https://doi.org/10.7554/eLife.63402.sa1
Author response https://doi.org/10.7554/eLife.63402.sa2

## Additional files

### Supplementary files

• Supplementary file 1. Primers used for PCR. Details for the primers used in qPCR are listed. Primers for the genes in which a 'Supplier' is listed were purchased commercially.

• Supplementary file 2. Sclerostin small-molecule inhibitors (SMIs) show no signs of hepatic toxicity. This table shows the results of liver function panels from peripheral blood drawn at the time of euthanasia for both rats after SQ implantation (4 wk) and rabbits after posterolateral spinal fusion (6 wk).

• Transparent reporting form

## Data availability

All data generated or analysed during this study are included in the manuscript and supporting file.

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
