## [Editor Report]

This manuscript examines the use of small molecule sclerotin inhibitors as alternatives to currently used biologicals such as BMP for interventions such as spinal fusion. The manuscript has important clinical significance and the strength of evidence is considered convincing.

---

## [Decision Letter]

**Decision letter after peer review:**

Thank you for submitting your article "Sclerostin Small Molecule Inhibitors Promote Osteogenesis by Activating Canonical Wnt and BMP Pathways" for consideration by *eLife*. Your article has been reviewed by 3 peer reviewers, including Carlos Isales as the Reviewing Editor and Reviewer #1, and the evaluation has been overseen by Mone Zaidi as the Senior Editor. The following individuals involved in review of your submission have agreed to reveal their identity: Fayez Safadi (Reviewer #2).

The reviewers have discussed the reviews with one another and the Reviewing Editor has drafted this decision to help you prepare a revised submission.

Summary:

In the manuscript by Sangadala et al. the authors examine the use of small molecule sclerotin inhibitors as alternatives for other biologicals currently used (BMP) for clinical approaches such as spinal fusion. The authors also examine the roles of the WNT and BMP pathways as mediators of this effect. Small molecule sclerotin inhibitors used were selected based on potency (F1, VA1 and C07) and were able to inhibit sclerostin/LRP5/6 interactions. in vivo effectiveness was demonstrated using ectopic mineralization in rats and spinal fusion rates in rabbits

Essential revisions:

1) The authors presented convincing data on identifying three potentials targets for inhibiting sclerostin function by regulating both Wnt and BMP2 pathways, the authors stated that these inhibitors are FDA approved. There is not information on these inhibitors in term what are they FDA approved for and the indications. This will clearly dictate how these inhibitors will affect bone metabolism and homeostasis. Hence, please add more information on each inhibitor in term of FDA approval indication, etc. The FDA-approved drugs used in the study need to be cited by their generic names.

2) As this paper is a drug discovery study, the data in Figure 2 need to derive avidity and/or affinity constants for each molecule. Figure 2B is an incomplete dose response, and the celling was not determined. This is important because the highest concentration (10um) was used in the subsequent experiments, but this dose was not determined to have the maximum effect.

3) Figure 4 should include a time course of the induced gene expression. It would be very helpful in the authors show the level of sclerostin mRNA or protein levels. It would also be helpful if the authors consider providing some evidence on osteoclast (TRAP staining from the in vivo experiment) or assessing the level of CTx in circulation. Given RANKL expression and production is highly expressed by osteocytes. Is there any assessment of osteocytes from the in vivo samples? If the osteoclast data is not available, then please provide more discussion on BMP2, SMI, Wnt, Slecrotsin cross-talk and osteoclasts.

4) In Figure 5, changes in mineralization assay are visually striking but would be helpful if this was also quantitated and presented in graph format, include number of times experiment done. Figure 5 could also include a time course to see if the SMIs accelerate mineralization.

5) In Figure 6, I’m curious if the inflammatory reaction differed between BMP and the analogs, either locally or systemically. For consistency, the 3D renderings of the C07 induced bone in Figure 6B should be presented without the blue background like VA1 and rhBMP-2.

6) Figure 7 does not use BMP as a positive control, how do those results compare? Any assessment of bone quality, what about biomechanics? Figure 7 should contain histology of the new bone formation.

7) The claim that "Sclerostin SMIs are non-toxic" needs to be supported with formal studies or removed.

---

## [Author Response]

Essential revisions:1) The authors presented convincing data on identifying three potentials targets for inhibiting sclerostin function by regulating both Wnt and BMP2 pathways, the authors stated that these inhibitors are FDA approved. There is not information on these inhibitors in term what are they FDA approved for and the indications. This will clearly dictate how these inhibitors will affect bone metabolism and homeostasis. Hence, please add more information on each inhibitor in term of FDA approval indication, etc. The FDA-approved drugs used in the study need to be cited by their generic names.

We have now included information on these inhibitors with respect to their original FDA approval and their original clinical indications (Table 1).

2) As this paper is a drug discovery study, the data in Figure 2 need to derive avidity and/or affinity constants for each molecule. Figure 2B is an incomplete dose response, and the celling was not determined. This is important because the highest concentration (10um) was used in the subsequent experiments, but this dose was not determined to have the maximum effect.

We did try to repeat the in vitro binding studies to find out if we could achieve more than 15-20% inhibition of Sclerostin-LRP interaction. We could not get desired levels of inhibition, however. This could be due to following reasons:

The repurposed drugs are not designed to maximally bind LRP. The *in silico* parameters only allow partial binding to target sites.The LRP protein is immobilized on a membrane blot for binding assay to minimize requirement for LRP protein amount. If we perform solution binding kinetics in larger volumes, it would be requiring the milligram quantities of binding proteins. Current protein amounts cost was around $1200 which will be more than $20k if we purchase recombinant proteins for solution binding efforts.Affinity constant determination is possible if we can cover at least four doses on both sides of an apparent IC50 value. However, we are not able to reach an inhibition beyond IC20 in our case.

3) Figure 4 should include a time course of the induced gene expression. It would be very helpful in the authors show the level of sclerostin mRNA or protein levels. It would also be helpful if the authors consider providing some evidence on osteoclast (TRAP staining from the in vivo experiment) or assessing the level of CTx in circulation. Given RANKL expression and production is highly expressed by osteocytes. Is there any assessment of osteocytes from the in vivo samples? If the osteoclast data is not available, then please provide more discussion on BMP2, SMI, Wnt, Slecrotsin cross-talk and osteoclasts.

We now provide evidence on how the sclerostin SMIs affect osteoclasts by providing a Western Blot of OPG and RANKL protein levels in the presence of the SMIs (Figure 6B) and a functional assay of actual bone resorption (Figure 6C). Further discussion is also provided where appropriate.

4) In Figure 5, changes in mineralization assay are visually striking but would be helpful if this was also quantitated and presented in graph format, include number of times experiment done. Figure 5 could also include a time course to see if the SMIs accelerate mineralization.

While we did not provide quantification due to time/budget constraints, we now, as requested, provide in Figure 5 a time course to see if the SMIs accelerate mineralization (5C). We also now provide a dose-response figure on mineralization as well (5D).

5) In Figure 6, Im curious if the inflammatory reaction differed between BMP and the analogs, either locally or systemically. For consistency, the 3D renderings of the C07 induced bone in Figure 6B should be presented without the blue background like VA1 and rhBMP-2.

The 3D renderings of the C07 induced bone are now presented without the blue background like VA1 and rhBMP-2.

6) Figure 7 does not use BMP as a positive control, how do those results compare? Any assessment of bone quality, what about biomechanics? Figure 7 should contain histology of the new bone formation.

As requested, Figure 7 now shows BMP as a positive control as well as histology of the explants.

7) The claim that "Sclerostin SMIs are non-toxic" needs to be supported with formal studies or removed.

All claims that "Sclerostin SMIs are non-toxic" have now been removed.